# CONSIGN: CONFORMAL SEGMENTATION INFORMED BY SPATIAL GROUPINGS VIA DECOMPOSITION

**Bruno Viti**
Department of Mathematics and Scientific Computing
University of Graz, AT
BioTechMed-Graz
`bruno.viti@uni-graz.at`

**Elias Karabelas**
Department of Mathematics and Scientific Computing
University of Graz, AT
BioTechMed-Graz
`elias.karabelas@uni-graz.at`

**Martin Holler**
IDea_Lab
University of Graz, AT
BioTechMed-Graz
`martin.holler@uni-graz.at`

## ABSTRACT

Most machine learning-based image segmentation models produce pixel-wise confidence scores that represent the model's predicted probability for each class label at every pixel. While this information can be particularly valuable in high-stakes domains such as medical imaging, these scores are heuristic in nature and do not constitute rigorous quantitative uncertainty estimates. Conformal prediction (CP) provides a principled framework for transforming heuristic confidence scores into statistically valid uncertainty estimates. However, applying CP directly to image segmentation ignores the spatial correlations between pixels, a fundamental characteristic of image data. This can result in overly conservative and less interpretable uncertainty estimates. To address this, we propose CONSIGN (*Conformal Segmentation Informed by Spatial Groupings via Decomposition*), a CP-based method that incorporates spatial correlations to improve uncertainty quantification in image segmentation. Our method generates meaningful prediction sets that come with user-specified, high-probability error guarantees. It is compatible with any pre-trained segmentation model capable of generating multiple sample outputs. We evaluate CONSIGN against two CP baselines across three medical imaging datasets and two COCO dataset subsets, using three different pre-trained segmentation models. Results demonstrate that accounting for spatial structure significantly improves performance across multiple metrics and enhances the quality of uncertainty estimates.

## 1 INTRODUCTION

In many real-world applications, predictive machine learning models are increasingly used to support critical decision-making processes. However, these models often operate under various sources of uncertainty, including noisy data and limited observations. As a result, it is essential not only to generate accurate predictions, but also to assess the reliability of these predictions. Uncertainty quantification (UQ) provides a systematic framework for evaluating and communicating the degree of confidence in model outputs, see Abdar et al. (2021) for a recent review. Specifically, as deep learning models increasingly dominate segmentation tasks due to their high accuracy, it becomes equally important to assess the confidence of these predictions through UQ.

Several UQ approaches have been proposed in recent years, including Bayesian and ensemble methods, see Abdar et al. (2021); Huang et al. (2024); Lambert et al. (2024) for detailed reviews. One type, Bayesian methods, includes Monte Carlo dropout techniques (Gal & Ghahramani, 2016; Kendall & Gal, 2017). These methods enable uncertainty estimation by applying dropout at test time and sampling multiple forward passes to approximate a posterior distribution. The second category

of UQ methods consists of Deep Ensemble Networks (Lakshminarayanan et al., 2017; Mehrtash et al., 2020). Ensemble methods estimate uncertainty by combining predictions from multiple independently trained models, capturing diverse hypotheses. Kohl et al. (2018), instead, proposed a different architecture that combines a U-Net (Ronneberger et al., 2015) with a Variational Autoencoder (VAE) (Kingma et al., 2019). Along the same line, (Baumgartner et al., 2019) proposed PhiSeg, and (Monteiro et al., 2020) introduced the Stochastic Segmentation Network.

A common limitation of the above-discussed methods is that they do not provide statistical guarantees regarding the reliability or coverage of their predicted uncertainty. In other words, while these approaches may produce plausible predictions of uncertainty, there is no statistical guarantee that the predicted uncertainty actually matches the true uncertainty associated with the model and the data distribution. Conformal Prediction (CP) (Lei & Wasserman, 2014; Papadopoulos et al., 2002; Vovk et al., 2005) is a statistical approach to uncertainty quantification that has recently seen a surge of interest within the machine learning community. Essentially, CP provides a principled way to transform informal or heuristic uncertainty measures into rigorous ones (Angelopoulos et al., 2023).

The general workflow of CP can be outlined as follows: First, we need a fixed pre-trained model $f$ that has been trained on a dataset $\mathcal{D}_{train}$. Usually, the model is required to have some heuristic notion of uncertainty that should be made rigorous. Next, the pre-trained model is evaluated and adapted on the basis of a calibration dataset $\mathcal{D}_{cal}$, in order to obtain a calibrated notion of uncertainty with coverage guarantees. Finally, the calibrated model can be evaluated on a new test dataset $\mathcal{D}_{test}$ for which the desired coverage guarantees hold. The main assumption for the latter to hold is that the calibration and test datasets are exchangeable, which is true, for instance, if they are independent and identically distributed ($i.i.d$). We are interested in a conformal prediction approach that outputs for each test image $X_{test}$, a set of predictions $\mathcal{C}(X_{test})$, with some pre-defined guarantees regarding the accuracy of those predictions. In particular, we want to leverage a specific area of CP, Conformal Risk Control (CRC), which provide guarantees of the form

$$\mathbb{E}\Big[\ell\big(\mathcal{C}(X_{test}), Y_{test}\big)\Big] \leq \alpha, \tag{1}$$

where $\ell$ is any bounded loss function that shrinks as $\mathcal{C}$ grows and $\alpha$ is an user-defined parameter such that $1 - \alpha$ is the desired confidence. In particular, during the calibration step we produce prediction sets $\mathcal{C}_\lambda(\cdot)$, where the parameter $\lambda$ encodes the level of conservativeness: the higher the $\lambda$ the larger the prediction sets. We are interested in finding the best parameter $\hat{\lambda}$ that guarantees equation 1. Given a calibration set $\{(X_i, Y_i)\}_{i=1}^n$, the guarantee can be achieved by the choice

$$\hat{\lambda} = \inf\left\{\lambda : \hat{R}(\lambda) \leq \alpha - \frac{B - \alpha}{n}\right\}, \tag{2}$$

where $B$ is the maximum of the loss function and $\hat{R}(\lambda) = \frac{1}{n}\sum_{i=1}^n \ell(\mathcal{C}_\lambda(X_i), Y_i)$ is the empirical risk. The definition of $\mathcal{C}(\cdot)$ is crucial, and the quality of the algorithm heavily depends on it. In standard segmentation approaches, prediction sets are defined as

$$\mathcal{C}_\lambda(X^{ij}) = \{l : f(X^{ij})_l \geq 1 - \lambda\}, \quad \lambda \in [0, 1], \tag{3}$$

meaning that each pixel instead of being a singleton (the $\arg\max$ of the softmax probabilities $f(X^{ij})$) is a set containing all the labels that have a softmax score greater then the threshold $1 - \lambda$.

Recent works have extended basic conformal prediction methods to quantify uncertainty in image segmentation tasks. Wundram et al. (2024) apply pixel-wise CP to different segmentation models and evaluate the performance for binary segmentation tasks, while in Mossina et al. (2024) they extended CRC to address the multi-class segmentation challenge by constructing pixel-wise prediction sets of the form defined in equation 3. Blot et al. (2025) instead, extended CRC towards group conditional risk control, again in a pixel-wise setting. Wieslander et al. (2020) were among the first to introduce pixel-wise CP in medical imaging, while Davenport (2024) extended the approach by making the nonconformity score dependent on the distance to the mask boundaries. Brunekreef et al. (2024) tried to overcome pixel-wise CP approaches introducing a method where non-conformity scores are aggregated over similar image regions. The method, however, relies on a custom calibration strategy that depends heavily on the characteristics of the data and task. Teng et al. (2023) proposed a feature-based CP using deep network representations, which, however, leads to a need of model internal information that might not be available. A recent contribution from Bereska et al. (2025) adapts prediction sets based on proximity to critical vascular structures in medical imaging.

Mossina & Friedrich (2025) introduced a novel approach based on morphological operations, which, however, is currently limited to binary segmentation. Finally, Liu et al. (2025) introduced SACP, a spatial-aware CP method where the scores are aggregated across neighborhood pixels.

In summary, most prior works, in particular those who are generically applicable to pre-trained segmentation models, construct the set-valued predictions $\mathcal{C}(\cdot)$ only for each pixel separately, disregarding spatial correlations within the image. Since the coverage guarantee still holds by the design of the conformal prediction method, this leads to an unnecessarily large size of the set-valued prediction; i.e., the set of possible labels for different pixel regions is larger than it would need to be, given the spatial dependence of pixels.

In this work, we address this issue by developing *Conformal Segmentation Informed by Spatial Groupings via Decomposition* (CONSIGN), a method to leverage spatial correlation for improved conformal prediction sets. Building upon techniques that exploit Singular Value Decomposition (SVD) to extract principal directions of uncertainty, as presented in Belhasin et al. (2023); Nehme et al. (2023) for image restoration, we propose a method that transforms any segmentation model capable of generating sample predictions – such as those using dropout, Bayesian modeling, or ensembles – into one that produces spatially-aware set predictions with formal coverage guarantees. To achieve this, we defined novel spatially-aware prediction sets and developed a corresponding calibration strategy tailored to their unique characteristics. Most notably, due to the fine-range property of segmentation, our method can provide rigorous uncertainty bounds while only relying on a rather low number of principle directions. To showcase the versatility of our method, we apply it to a range of pre-trained models. As we show via numerical experiments, the fact that our approach acknowledges spatial correlations in the segmentation masks, allows us to produce much tighter and more meaningful set-valued predictions compared to a direct pixel-wise approach that does not account for spatial correlations, see Figure 1 for an example.

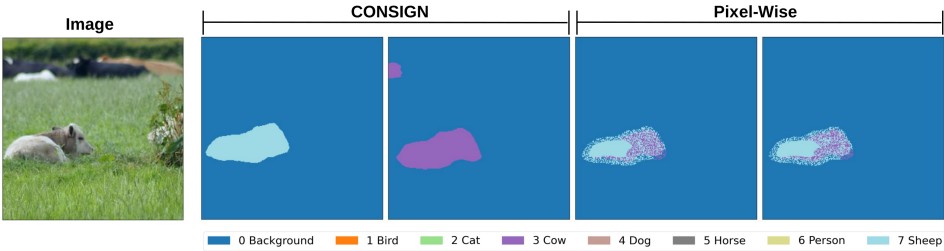

Figure 1: Images sampled from the prediction set $\mathcal{C}^*$ of CONSIGN, and from the pixel-wise one $\mathcal{C}^{PW}$. The pixel-wise method treats each pixel independently, disregarding correlation between pixels and including inconsistent predictions in the prediction set $\mathcal{C}^{PW}$. In contrast, our method samples masks by sampling different weights for the principal components extracted via the SVD approach, which captures spatial and contextual dependencies. This results in more realistic and coherent segmentations, as seen in the smooth transition between the sheep and cow in our method's results.

## 2 METHODS

### 2.1 PROBLEM DEFINITION

We want to develop a method that provides meaningful statistically valid guarantees for predictions of segmentation models that take into account spatial correlations. That is, instead of having a model $f$, trained on $\{(X_i, Y_i)\}_{i=1}^{N_{tr}}$, which outputs a single prediction $f(X_{test})$, we want a set of predictions $\mathcal{C}_\lambda(X_{test})$ such that equation equation 1 holds. The set of predictions depends on a parameter $\lambda$ that is calibrated using a calibration set $\{(X_i, Y_i)\}_{i=1}^{N_{cal}}$, disjoint from the training one. In our case, $X \in \mathbb{R}^{W \times H \times C}$ are images while $Y \in \{1, \ldots, L\}^{W \times H}$ are the corresponding segmentations. Moreover, we consider models $f : \mathbb{R}^{W \times H \times C} \to \mathbb{R}^{W \times H \times L}$, which take in input an image and give as output softmax probabilities for $L$ labels.

Our method consists of two main components: The construction of a spatially-aware Set and the Calibration. In the first step, we identify the uncertain regions and create a new basis of vectors $\{\mathbf{u}_i\}_{i=1}^K$ that characterizes these areas. Those vectors are the key component for the construction of our prediction set $\mathcal{C}_\lambda$. In the calibration step, we find the best parameter $\hat{\lambda}$ and corresponding prediction set $\mathcal{C}_{\hat{\lambda}} \subset \{1, \ldots, L\}^{W \times H}$ that satisfies equation 1.

## 2.2 CONSTRUCTION OF SPATIALLY-AWARE SET

We want to leverage the correlation between pixels to provide more meaningful and precise uncertainty regions. In particular, we are interested in constructing a set $\mathcal{C}_\lambda(\cdot)$ that contains predictions whose uncertain pixels change jointly, following a meaningful structure, rather than independently. To this end, principal component analysis (PCA) provides a framework for capturing and representing these joint variations in uncertainty. For our purpose, a PCA approach can make use of the different samples obtained from a pre-trained model $f$ to gain insights into both the location of the uncertain regions and the correlation between pixels within those regions. This information is crucial for enhancing the interpretability of the model.

This concept of extracting uncertainty regions using principal components is one of the strengths of our model and has shown its effectiveness in previous works on image regression such as Belhasin et al. (2023); Nehme et al. (2023).

First, we identify the uncertain regions using a pre-trained model $f$. For this, our method does not require any particular model $f$, as long as the model can be used to generate samples $\hat{\mathbf{s}}_1, \ldots, \hat{\mathbf{s}}_{N_s} \in \mathbb{R}^{WHL}$ that correspond to heuristic uncertainties like softmax scores. Following the main idea of Belhasin et al. (2023), we construct a sample matrix $\hat{S}(X) = [\hat{\mathbf{s}}_1, \ldots, \hat{\mathbf{s}}_{N_s}]$, compute its mean and extract the uncertain regions through an SVD (Golub & Reinsch, 1971) as

$$\boldsymbol{\mu}(X) = \frac{1}{N_s} \sum_{n=1}^{N_s} \hat{\mathbf{s}}_n, \quad \hat{S} - \boldsymbol{\mu}(X) \cdot \mathbf{1}_{N_s}^T = U\Sigma V^T. \tag{4}$$

Note that here, as detailed below, it is sufficient to use a reduced SVD and only compute the first $K < \min\{WHL, N_s\}$ singular values, with $K \in \{2, 5\}$ in our experiments. Each column $\mathbf{u}_k \in \mathbb{R}^{WHL}$ of $U$ is a basis vector for the space of samples, aligned with the directions of maximum variance in the data. Building on this interpretation, to construct our prediction set, we first compute quantiles of the coefficients of the basis vectors $\mathbf{u}_k$, $k = 1, \ldots, K$, over the $N_s$ samples via

$$a_k = \mathcal{Q}_{\frac{\alpha}{2}}\left(\{\langle \mathbf{u}_k, \hat{\mathbf{s}}_n - \boldsymbol{\mu}(X)\rangle\}_{n=1}^{N_s}\right), \ b_k = \mathcal{Q}_{1-\frac{\alpha}{2}}\left(\{\langle \mathbf{u}_k, \hat{\mathbf{s}}_n - \boldsymbol{\mu}(X)\rangle\}_{n=1}^{N_s}\right). \tag{5}$$

Here $\mathcal{Q}_\alpha(\cdot)$ is the $\alpha-$quantile and $\langle \cdot, \cdot \rangle$ the scalar product. Then, we define bounds for the basis coefficients as

$$A_k = \frac{a_k + b_k}{2} - \lambda \Sigma_{k,k} \frac{b_k - a_k}{2}, \ B_k = \frac{a_k + b_k}{2} + \lambda \Sigma_{k,k} \frac{b_k - a_k}{2}. \tag{6}$$

The derivation of equation 6 is heuristic and is designed to obtain symmetric bounds around the empirical quantile midpoint that scale linearly with $\lambda$. The weighting by $\Sigma_{k,k}$ (the k-th singular value) creates wider bounds for principal components with higher variance (larger $\Sigma_{k,k}$), allowing more flexibility where the model shows more uncertainty. The parameter $\lambda$ will either shrink or enlarge the bounds, and it will be calibrated in the calibration step in order to fulfill equation equation 1. Let $Y = P(\boldsymbol{\sigma})$ represent the predicted labels for an element $\boldsymbol{\sigma} \in \mathbb{R}^{WHL}$, with $P$ beeing the argmax applied along the label dimension of the reshaped element $\boldsymbol{\sigma} \in \mathbb{R}^{W \times H \times L}$. With this, we define the prediction set

$$\mathcal{C}_\lambda^{*-}(X) = \left\{ Y \ : \ \exists \mathbf{c} \in \underset{k=1}{\overset{K}{\times}} [A_k(X), B_k(X)] : Y = P\left(\boldsymbol{\mu}(X) + \sum_{k=1}^K c_k \mathbf{u}_k(X)\right) \right\}.$$

Note that this set contains the predictions such that there exists a score vector $\boldsymbol{\sigma} \in \mathbb{R}^{WHL}$ whose deviation from the mean can be expressed as linear combination of the first $K$ basis vectors $\mathbf{u}_1, \ldots, \mathbf{u}_K$ with coefficients $c_k$ inside of the bounds defined in equation 6. A big advantage of our approach here is that, as we will see in the experiments, meaningful prediction sets of this form can already be obtained with $K \in \{2, 5\}$. This significantly reduces the computational load compared to a full

basis representation with $K = WHL$, and is also a major difference to the regression approach of Belhasin et al. (2023): Since our method includes a nonlinear quantization-type step $P(\boldsymbol{\sigma})$, mapping softmax outputs $\boldsymbol{\sigma}$ to discrete predicted labels $Y$, we can enforce the coefficients of most of the basis vectors $\mathbf{u}_{K+1}, \ldots, \mathbf{u}_{WHL}$ to be zero and still be able to reconstruct the ground truth for some combination of coefficients $c_k$. In other words, even with a truncated PCA the prediction set will generally include the ground truth provided that $\lambda$ is large enough (with pathological exceptions, which we discuss in the next section). In contrast, in the regression approach of Belhasin et al. (2023), the authors need to introduce a special procedure to achieve coverage guarantees also for a truncated PCA. Independent of this, we still allow for a user-defined accuracy rate $\beta$ in our prediction set as follows: We say that two predictions $Y_1$ and $Y_2$ coincide for a label-wise accuracy rate of $\beta$ and write $Y_1 \stackrel{\beta}{=} Y_2$ if $\frac{1}{L} \sum_{l=1}^{L} \frac{\sum_{ij} \mathbb{I}(Y_1^{ij}=l \wedge Y_2^{ij}=l)}{\sum_{ij} \mathbb{I}(Y_1^{ij}=l)} > \beta$, where $\beta$ is a second user-defined parameter that control the desired accuracy and $i, j$ are pixel coordinates. Notably, we adapt the standard accuracy rate from regression to enforce a label-wise accuracy rate, thereby avoiding a bias towards more frequent labels. Using this, we now define the final prediction set and relative loss function as

$$\mathcal{C}_\lambda^*(X) = \left\{ Y \ : \ \exists \mathbf{c} \in \underset{k=1}{\overset{K}{\times}} [A_k(X), B_k(X)] : Y \stackrel{\beta}{=} P\left(\boldsymbol{\mu}(X) + \sum_{k=1}^{K} c_k \mathbf{u}_k(X)\right) \right\} \tag{7}$$

$$\ell(\mathcal{C}_\lambda^*(X), Y) = 1 - \mathbb{I}_{\mathcal{C}_\lambda^*(X)}(Y), \tag{8}$$

which is a bounded loss function that shrinks as $\lambda$ increases. Here, and trough all the paper, $\mathbb{I}_{\mathcal{C}(X)}(Y)$ refers to the indicator function of set $\mathcal{C}(X)$, i.e. equal to 1 if $Y \in \mathcal{C}(X)$, 0 otherwise.

## 2.3 CALIBRATION

Having defined the $\lambda$-dependent prediction set $\mathcal{C}_\lambda^*(X)$ as in equation 7, we can in theory use the standard calibration procedure of Angelopoulos et al. (2024) to obtain a coverage of the form

$$\mathbb{E}[\ell(\mathcal{C}_\lambda^*(X_{test}), Y_{test})] = \mathbb{P}[Y_{\text{test}} \notin \mathcal{C}_\lambda^*(X_{\text{test}})] \leq \alpha. \tag{9}$$

This standard calibration procedure iterates trough the calibration set, evaluates the empirical loss $\hat{R}(\lambda) = \frac{1}{n} \sum_{i=1}^{n} \ell(\mathcal{C}_\lambda^*(X_i), Y_i)$ and checks if $\hat{R}(\lambda) \leq \alpha - \frac{1-\alpha}{n}$ ($B = 1$ in our case). If the empirical loss is above this threshold, the procedure is repeated with an increased $\lambda$. Otherwise, $\lambda = \lambda^\dagger$ is calibrated and the desired coverage for a new test point can be guaranteed. For example, if $\alpha = 0.1$, and $n = 100$, the procedure searches for $\lambda^\dagger$ such that more than $90.9\%$ of the calibration points $(X_i, Y_i)$ satisfy $Y_i \in \mathcal{C}_{\lambda^\dagger}^*(X_i)$. In practice, however, we need to adapt this procedure, as the rather involved form of our prediction set does not allow us to easily check if $Y \in \mathcal{C}_\lambda^*(X)$. In fact, exhaustively checking all possible values of $\mathbf{c}$ is computationally infeasible. To address this, we formulate and numerically solve a constrained minimization problem such as

$$\mathbf{c}^* = \arg\min_{\mathbf{c} \in \mathcal{B}} \mathcal{L}(Y, P(\boldsymbol{\mu}(X) + \sum_{k=1}^{K} c_k \mathbf{u}_k)), \quad \mathcal{B} = \underset{k=1}{\overset{K}{\times}} [A_k(X), B_k(X)] \tag{10}$$

$$\mathcal{L}(Y, P(\boldsymbol{\sigma})) = 1 - \frac{1}{L} \sum_{l=1}^{L} \frac{\sum_{ij} \mathbb{I}(Y^{ij} = l \wedge P(\boldsymbol{\sigma})^{ij} = l)}{\sum_{ij} \mathbb{I}(Y^{ij} = l)} \tag{11}$$

If the numerical solution $\mathbf{c}^* \in \mathcal{B}$ satisfies $Y \stackrel{\beta}{=} P(\boldsymbol{\mu}(X) + \sum_{k=1}^{K} c_k^* \mathbf{u}_k(X))$, then we can guarantee that $Y \in \mathcal{C}_\lambda^*(X)$. However, due to the numerical nature of the optimization process, which is described in Algorithm 2, global optimality cannot be guaranteed. In practice, this means that even if a suitable $\mathbf{c}$ exists within the current bounds, the solver may fail to find it, possibly leading to an unnecessary increase in $\lambda$. Despite this, the statistical guarantee of the overall algorithm is not compromised. Even when $\lambda$ is increased beyond what is strictly necessary, the method will still output a valid $\hat{\lambda}$ that guarantees equation 9. The only consequence is that the resulting bounds on $\mathbf{c}$ may be more conservative. Moreover, since previously accepted $\mathbf{c}$ remain valid under expanded bounds, and additional segmentations $Y$ may be in $\mathcal{C}_\lambda^*(X)$ as the bound increases, the loss equation 8 is guaranteed to be non-increasing. A summary of the resulting procedure is sketched in Algorithm 1, and the following lemma (which is a direct consequence of Angelopoulos et al. (2024, Theorem 1) and proven in Appendix A) provides the resulting coverage guarantees for this algorithm.

**Lemma 1.** *If Algorithm 1 terminates with $\hat{\lambda} < \infty$, and if the $(X_1, Y_1), \dots (X_n, Y_n)$ used in this algorithm are exchangeable with $(X_{test}, Y_{test})$, then*

$$\mathbb{P}\left[ Y_{test} \in C^*_{\hat{\lambda}}(X_{test}) \right] \geq 1 - \alpha.$$

### 2.3.1 Termination of the algorithm

As mentioned in Section 2.2, there may be pathological cases—such as zero or poorly aligned entries in the principal directions $\mathbf{u}_k$—in which our method cannot converge even as $\lambda \to \infty$. This behavior is actually desirable, as it prevents the method from silently producing prediction sets that fail to satisfy the intended coverage guarantees. In practice, we impose a maximum value of $\lambda$ and stop the algorithm if it reaches $\lambda_{max}$, so that the user is explicitly informed when no useful or informative prediction set can be obtained for the chosen parameters $\alpha$ and $\beta$.

---

**Algorithm 1** Calibration algorithm for CONSIGN

**Input:** $\alpha, \beta, d\lambda, \{(X_i, Y_i)\}_{i=1}^{N_{cal}}$     **Output:** $\hat{\lambda}$

1: **pre-compute:** $\{(\boldsymbol{\mu}(X_i), \hat{S}^i, U^i, \Sigma^i)\}_{i=1}^{N_{cal}}, \{\{(a_k^i, b_k^i)\}_{k=1}^K\}_{i=1}^{N_{cal}}$ as in equation 4, equation 5

2:  $\lambda \leftarrow 0; \hat{R} \leftarrow 1; \mathcal{I} \leftarrow \emptyset$

3: **while** $\hat{R} > \alpha - \frac{1-\alpha}{N_{cal}}$ **do**

4:      **for** $i \leftarrow 1$ **to** $N_{cal} \setminus \mathcal{I}$ **do**

5:          $\left(A_k, B_k\right) \leftarrow \left( \frac{a_k^i + b_k^i}{2} - \lambda \Sigma_{k,k}^i \frac{b_k^i - a_k^i}{2}, \frac{a_k^i + b_k^i}{2} + \lambda \Sigma_{k,k}^i \frac{b_k^i - a_k^i}{2} \right)$                 $\triangleright$ for each $k$

6:          $\mathcal{B} \leftarrow \times_{k=1}^K [A_k, B_k]$

7:          $\mathbf{c}^* \leftarrow \text{approx\_solver}(\arg\min_{\mathbf{c} \in \mathcal{B}} \mathcal{L}(Y_i, P(\boldsymbol{\mu}(X_i) + \sum_{k=1}^K c_k \mathbf{u}_k^i)))$   $\triangleright$ using e.g. Alg. 2

8:          $\boldsymbol{\sigma} \leftarrow \boldsymbol{\mu}(X_i) + \sum_{k=1}^K c_k^* \mathbf{u}_k^i$

9:          **if** $Y_i \overset{\beta}{=} P(\boldsymbol{\sigma})$ **then** $\mathcal{I} \leftarrow \mathcal{I} \bigcup \{i\}$

10:      $\hat{R} \leftarrow 1 - \frac{|\mathcal{I}|}{N_{cal}}$

11:      **if** $\hat{R} \leq \alpha - \frac{1-\alpha}{N_{cal}}$ **then** $\hat{\lambda} \leftarrow \lambda$ **else** $\lambda \leftarrow \lambda + d\lambda$

---

## 3 Experiments

We proved that our method creates prediction sets containing the ground truth with user-defined guarantees. Now we validate the method numerically, showing its performances through different experiments and metrics. In particular, we are interested in showing that our method provides prediction sets with lower uncertainty volume compared to the pixel-wise baseline defined in Angelopoulos et al. (2020), and the spatial-aware method SACP used in Liu et al. (2025). We define the uncertainty volume as the number of predictions in a prediction set $\mathcal{C}_\lambda$. With equal theoretical guarantees, we aim for the method that has a lower volume. In the next section, we quantify the volume and show that our method reliably produces a smaller estimate.

### 3.1 Datasets and Baselines

We are interested in applying our method to different datasets and pre-trained models $f$, to show its effectiveness regardless of the setting. We use three medical datasets (M&Ms-2Campello et al. (2021); Martín-Isla et al. (2023), MS-CMR19Gao et al. (2023); Wu & Zhuang (2022); Zhuang (2018), LIDCArmato III et al. (2015)), and two subsets of the COCO dataset Lin et al. (2014). For each dataset we use a different model $f$, in order to show flexibility of our approach also with respect to the segmentation model. For the two cardiac datasets M&Ms-2 and MS-CMR19, we produce samples through a U-Net Ronneberger et al. (2015) trained with dropout. For the LIDC dataset, we employ the method proposed by Kohl et al. (2018), which intrinsically contains a way of generating different samples. Finally, for the subsets of the COCO dataset, we employ a ensemble networks strategy based on DeepLabV3+ Chen et al. (2017; 2018) and generate different samples using different backbones. See Appendix B for further details on datasets and pre-trained models.

As pixel-wise (PW) baseline, we use RAPS (Angelopoulos et al., 2020), a CP method that forms prediction sets by including labels until the cumulative softmax sum plus a regularization term exceeds $\lambda$. Let $\pi$ be a permutation of indices such that $f(X^{ij})_{\pi(1)} \geq \cdots \geq f(X^{ij})_{\pi(L)}$, then

$$\mathcal{T}^{PW}(X^{ij}) = \{\pi(1), \ldots, \pi(k)\}, \quad k = \min \left\{ l \in \{1, \ldots, L\} : \sum_{m=1}^{l} f(X^{ij})_{\pi(m)} + r(l) > \lambda \right\}. \tag{12}$$

The term $r(l)$ is defined as $\theta \cdot (o(l) - k_{reg})^+$ where $\theta$ and $k_{reg}$ are hyperparameter, while $o(l)$ is the ranking of $l$ among the label based on the probabilities $\pi$. See Appendix D for details. Based on this, a pixel-wise prediction set $\mathcal{C}_\lambda^{PW-}$ and a relaxed version with $\beta$ accuracy $\mathcal{C}_\lambda^{PW}$ are defined as

$$\mathcal{C}_\lambda^{PW-}(X) = \left\{ Y : \forall i,j \ Y^{ij} \in \mathcal{T}^{PW}(X^{ij}) \right\}, \quad \mathcal{C}_\lambda^{PW}(X) = \left\{ Y : \exists \tilde{Y} \in \mathcal{C}_\lambda^{PW-}(X) : Y \stackrel{\beta}{=} \tilde{Y} \right\}. \tag{13}$$

For comparison with spatial-aware approaches, we employ SACP (Liu et al., 2025), where the pixel-wise cumulative sums of softmax - used to construct the prediction sets - are aggregated over local neighborhoods. Similarly to RAPS, $\mathcal{T}^{SACP}$ is defined as the set of labels whose cumulative scores plus regularization, after aggregation over local neighborhoods, exceed $\lambda$. Given $\mathcal{T}^{SACP}$, we can define the corresponding $\mathcal{C}_\lambda^{SACP-}$ and $\mathcal{C}_\lambda^{SACP}$ as in equation 13. See Appendix B for further details. The baselines calibration algorithms have a similar structure as Algorithm 1, only that the loss function is given as $\ell(\mathcal{C}_\lambda^{PW/SACP}(X), Y) = 1 - \mathbb{I}_{\mathcal{C}_\lambda^{PW/SACP}(X)}(Y)$ and that one can directly check if a ground-truth is contained in the prediction set. See Algorithm 3 for details.

## 3.2 SAMPLING AND METRICS

In order to quantitatively compare our approach to the baselines, we want to compute the volume of the prediction sets, which in this case is given as the number of different segmented images contained in this set. Since the definition and value of the accuracy rate $\beta$ is the same for both methods, we focus on the volume of the prediction sets $\mathcal{C}^{*-}(X), \mathcal{C}^{SACP-}(X)$ and $\mathcal{C}^{PW-}(X)$. For the pixel-wise method, this volume is given explicitly as $|\mathcal{C}^{PW-}(X)| = \prod_{i,j=1}^{W,H} |\mathcal{T}^{PW}(X^{ij})|$, and analogously for SACP. For $\mathcal{C}^{*-}(X)$, however, we can only estimate this volume using sampling, and in order to have a fair comparison, we use the same estimates for both methods. Recall that the sampling from our method means sampling coefficients $\mathbf{c} \in \mathcal{B}$ and generating the resulting segmentations as in equation 7, while for the baselines, it means to sample independently for each pixel $(i, j)$ possible labels contained in $\mathcal{T}^{PW}(X^{ij})$ and $\mathcal{T}^{SACP}(X^{ij})$. Using this sampling, define

$$\mathcal{Y}^*(X) = \{\hat{Y}_s\}_{s=1}^S \quad \mathcal{Y}^{PW}(X) = \{\hat{Y}_s\}_{s=1}^S, \quad \text{and} \quad \mathcal{Y}^{SACP}(X) = \{\hat{Y}_s\}_{s=1}^S,$$

to be samples sets from $\mathcal{C}^{*-}(X), \mathcal{C}^{PW-}(X), \mathcal{C}^{SACP-}(X)$, respectively, for a given test point $X$.
We evaluate our method and the baselines across three different metrics: Chao estimator (Chao, 1984), Sample-based Estimated Coverage ($sEC$) and correlation. The first metric is taken from the species richness estimation problem, which tries to estimate the true number of species based on sample data. In our setting, we aim to estimate the number of unique segmentations contained in a prediction set. The estimator provides a lower bound for the true number of segmentation and is defined as

$$\hat{N}_{CH}(\mathcal{Y}) := S_1 + \frac{f_1^2}{2f_2},$$

where $S_1$ is the number of unique samples, $f_1$ is the number of vectors sampled exactly once and $f_2$ is the number of vectors sampled exactly twice. The number of unique segmentations can grow rapidly. Therefore, we evaluate the estimator and study its behavior for different sample sizes $S$. The second metric, tries to estimate the volume of uncertainty through the empirical coverage. The empirical coverage $EC = \frac{1}{N_{test}} \sum_{i=1}^{N_{test}} \mathbb{I}_{\mathcal{C}_{\hat{\lambda}}(X_i)}(Y_i)$, should be, on average, greater or equal than $1 - \alpha$. However, while we can evaluate the empirical coverage for the baselines methods, we can only estimate it for our method since we do not know the best coefficient $\mathbf{c}$. Therefore, we introduce the Sample-based Estimated Coverage as

$$sEC(\mathcal{Y}) := \frac{1}{N_{test}} \sum_{i=1}^{N_{test}} \min \left( 1, \sum_{\hat{Y}_s \in \mathcal{Y}} \mathbb{I}(Y_i \stackrel{\beta}{=} \hat{Y}_s) \right).$$

This metric should converge to $1 - \alpha$ for an increasing number of random samples, and a faster convergence suggests that the corresponding set occupies a lower-dimensional subspace. The last metric is the averaged Pearson correlation

$$\hat{\rho}(\mathcal{Y}) := \frac{2}{S(S-1)} \sum_i \sum_{j>i} |\rho_{ij}(\hat{Y}_i, \hat{Y}_j)|,$$

which can be useful to quantify how much different random segmentation are correlated. A strong internal correlation also indicates that the samples are confined to a lower-dimensional manifold. In contrast, the near-independence of samples suggests a higher intrinsic dimensionality. See Appendix C for further details on the metrics.

## 3.3 RESULTS

We compare the baselines with our CONSIGN approach using two different numbers of principal components, specifically $K = 2$ and $K = 5$. We evaluate the metrics across five random calibration/test splits, and we use different combinations of parameters $\alpha$ and $\beta$ depending on the datasets and the pre-trained model. For safety-critical applications, like medical imaging, smaller values of $\alpha$ (e.g $\alpha = 0.05$) are preferred to enforce stricter confidence requirements. In contrast, exploratory analyses can tolerate larger $\alpha$ (e.g $\alpha = 0.3$). The parameter $\beta$ controls the acceptable pixel-wise accuracy: medical datasets with fine, small-scale structures benefit from higher $\beta$ (e.g $\beta > 0.8$). Whereas segmenting large, distinct objects may allow lower $\beta$ while still achieving reliable UQ. Moreover, optimal choices of $\alpha$ and $\beta$ also depend on the capability of the pre-trained model: stronger models permit more liberal choices, yielding smaller prediction sets without sacrificing reliability, whereas weaker models require more conservative settings. In the following figures, error bars will refer to the standard deviation across the five random splits and will depict $\pm 1$ standard deviation.

In Figure 2, we present the Chao estimator for various sample sizes. The estimator for CONSIGN is consistently bounded by the baselines estimators, indicating a smaller volume of uncertainty. In the LIDC experiment, the difference between the methods is maximized, with several orders of magnitude separating the estimators. In contrast, during the COCO experiments, the high uncertainty results in our method also producing a high Chao estimator. In the case of $K = 5$, the model has access to more principal components, which gives the coefficients $\mathbf{c}$ greater flexibility. The higher degree of freedom leads to a wider range of possible predictions and a higher estimated value of $\hat{N}_{CH}$. Moreover, CONSIGN's performance depends on the sample quality from the pre-trained model. With the probabilistic U-net, which is designed to produce meaningful samples, results improve by orders of magnitude, but using dropout for sampling leads to less pronounced gains over baselines. In Figure 3, we show the behavior of the $sEC$. CONSIGN, owing to its spatial

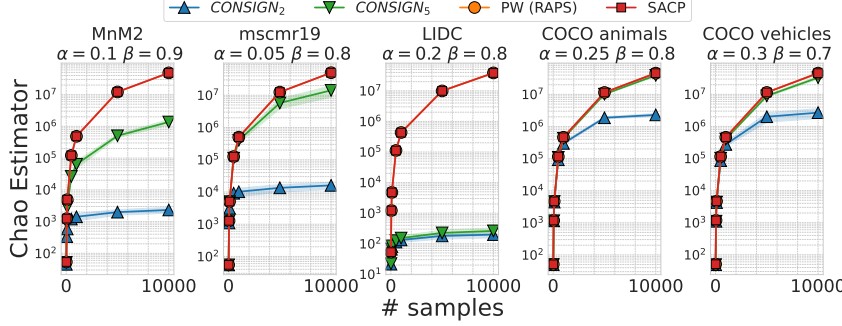

Figure 2: $\hat{N}_{CH}(\mathcal{Y}^*)$, $\hat{N}_{CH}(\mathcal{Y}^{PW})$ and $\hat{N}_{CH}(\mathcal{Y}^{SACP})$. Larger values indicate larger prediction set

awareness, outperforms RAPS and SACP by achieving empirical coverage with fewer samples. In the COCO-vehicle experiment, even a small sample of ten predictions meets the user-defined coverage requirement, indicating greater efficiency and precision in capturing relevant segmentations. Finally, in Figure 4, we compare the correlation between predictions sampled from $\mathcal{Y}^*$, $\mathcal{Y}^{SACP}$

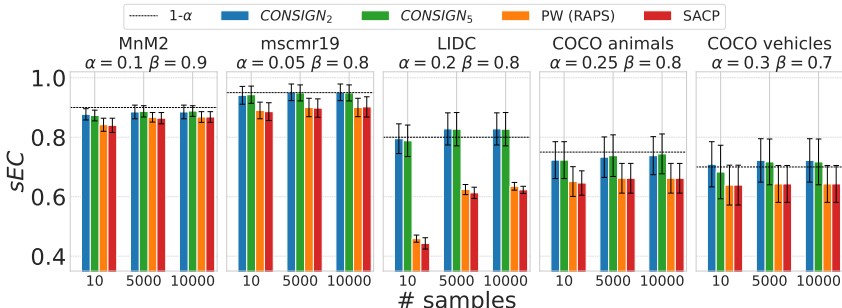

Figure 3: $sEC(\mathcal{Y}^*)$, $sEC(\mathcal{Y}^{PW})$ and $sEC(\mathcal{Y}^{SACP})$. Values close to $1-\alpha$ indicate better coverage

and $\mathcal{Y}^{PW}$. By using a linear combination of principal components to construct predictions, we enhance the consistency of our predictions in correlated regions, resulting in higher correlation among them. Moreover, CONSIGN exhibits a monotonic increase in correlation between samples, indicating consistency in capturing spatial structure. In contrast, the baselines show a decreasing trend in correlation, suggesting that the samples become nearly independent and fail to reflect any coherent shared structure. It can also visually observed that the accounting of spatial correlations, as done in CONSIGN, leads to a more meaningful set of possible segmentations: In Figure 5, we show how our method smoothly transitions between classes, jointly modifying regions that are uncertain and highly correlated. While the visualizations of pixel-wise samples may appear unrealistic, this behavior is expected, as pixel-wise prediction sets contain all combinatorial label configurations rather than structured samples.

In general, we can observe that the results for SACP are comparable to the pixel-wise ones. Aggregating softmax scores across pixels is mainly a post-processing step; however, a similar aggregation happens implicitly during the training of models $f$. Relying solely on this additional step does not effectively capture the true spatial correlations and results in a method that is comparable to a pixel-wise approach. In contrast, our method explicitly identifies correlated regions, outperforming the baselines across all metrics, proving an advantage in reducing the volume of uncertainty while providing consistent and qualitatively superior predictions.

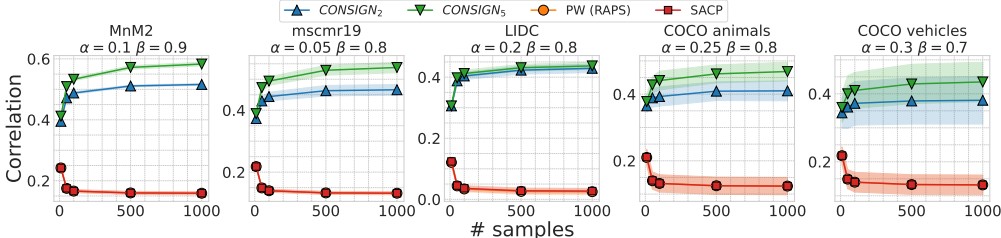

Figure 4: $\hat{\rho}(\mathcal{Y}^*)$, $\hat{\rho}(\mathcal{Y}^{PW})$ and $\hat{\rho}(\mathcal{Y}^{SACP})$. Larger values indicate greater spatial correlation

## 4 CONCLUSIONS AND LIMITATIONS

We developed a method that transforms heuristic and overconfident softmax scores into predictions backed by user-defined statistical guarantees. We exploit SVD techniques from previous approaches, such as Belhasin et al. (2023); Nehme et al. (2023), to introduce a new spatially-aware conformal prediction approach for image segmentation. Our approach stands out for three main reasons: First, we harness the power of spatial correlation to significantly improve segmentation quality while minimizing uncertainty. This results in a robust tool that allows users to sample insightful predictions with solid statistical assurances. Second, our method is easily applicable to any segmentation model that offers samples of predictions. Finally, by exploiting the classification nature of our setting and

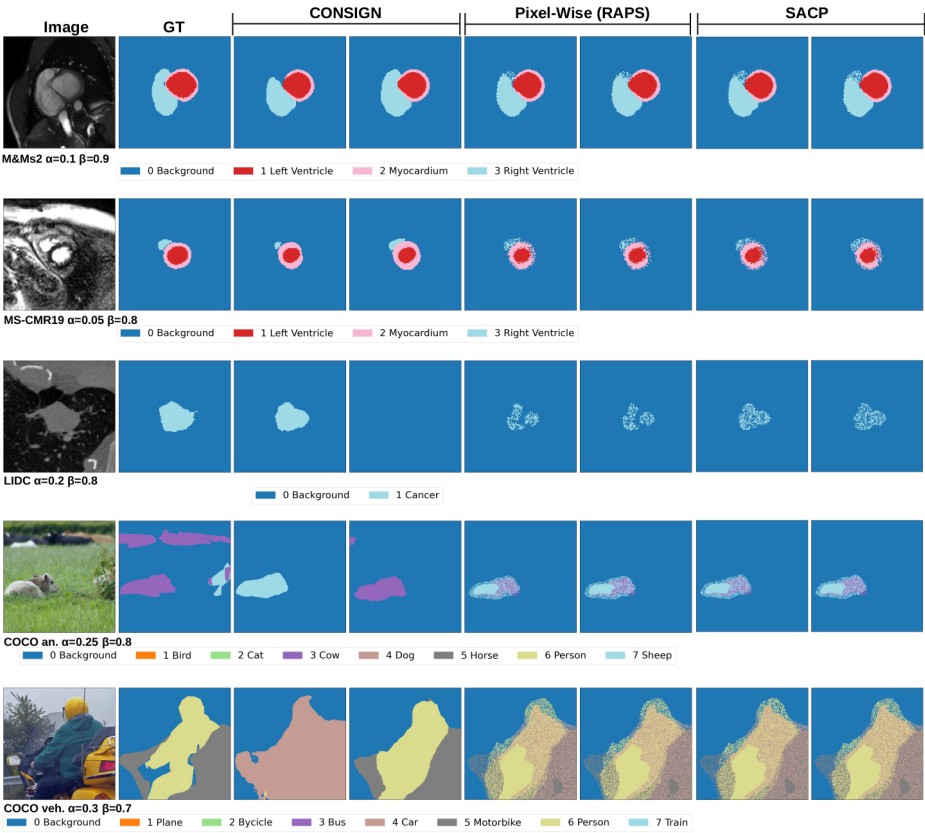

Figure 5: Qualitative comparison between samples from $\mathcal{Y}^*$ $(K = 5)$, $\mathcal{Y}^{PW}$ and $\mathcal{Y}^{SACP}$.

the non-linear projection $P(\cdot)$, we were able to reformulate the theory of Belhasin et al. (2023) in a way that yields more interpretable and practically meaningful bounds.

Currently, main limitations of our method are as follows: As with any (standard) conformal-prediction based method, the guarantees hold true under exchangeability assumptions on the data. Distribution-shifts or out-of distribution data are currently not addressed by our method. Extensions of conformal prediction in this direct exist, see e.g. Gibbs & Candes (2021), and a future goal is to extend our method in this direction. A second limitation of our method is the implicit form of the prediction set $\mathcal{C}^*_{\hat{\lambda}}(X)$, which increases the computational cost, see Appendix D for details, and makes it numerically challenging to evaluate if a given candidate segmentation is in $\mathcal{C}^*_{\hat{\lambda}}(X)$ or not. Nevertheless, we believe that this is not a major issue, since the online generation and sampling from the prediction set, which is the main application of our method, is still comparably fast.

## ACKNOWLEDGMENTS

The authors acknowledge the National Cancer Institute and the Foundation for the National Institutes of Health, and their critical role in the creation of the free publicly available LIDC/IDRI Database used in this study.

This research was funded in whole or in part by the Austrian Science Fund (FWF) 10.55776/F100800. B. V. and E. K. acknowledge funding from BioTechMed-Graz Young Research Group Grant CICLOPS.

## LLMS DECLARATION

The authors utilized large language models (LLMs) to check grammar and rewrite sentences.

REPRODUCIBILITY STATEMENT

In Appendix A we give the main proof of our method.

The calibration algorithm for CONSIGN is described in Algorithm 1, while the calibration algorithm of the pixel-wise method and SACP are described in 3. The optimization algorithm is described in Table 2, while Tables 3-4-5 provide the calibrated $\hat{\lambda}$ for all the experiments.

In Appendix B we describe the datasets, providing training/calibration/test splitting and references to download the datasets, which are all publicly available. We provide external links for the pre-trained models, and we discuss the implementation of the U-Net. Finally in Appendix D we discuss the hyper-parameters and hardware specifics.

Moreover, consistent with ICLR guidelines, we plan to share an anonymous repository with reviewers and ACs during the discussion phase, and will make the code public upon acceptance. The code is available at `https://github.com/onurbbruno/CONSIGN`.

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

## A PROOF OF LEMMA 1.

*Proof.* This is a direct consequence of (Angelopoulos et al., 2024, Theorem 1): It is clear that the loss $\lambda \mapsto L_{X,Y}(\lambda) := 1 - \mathbb{I}(Y \in C_\lambda(X))$ is non-increasing for all $(X, Y)$. Further, with a finite $\hat{\lambda}$ as provided by our algorithm, it is clear that

$$\infty > \hat{\lambda} \geq \lambda^\dagger := \inf \left\{ \lambda \; : \; \frac{1}{n+1} \left( \sum_{n=1}^{n} L_{X_i,Y_i}(\lambda) + 1 \right) \leq \alpha \right\}.$$

By (Angelopoulos et al., 2024, Theorem 1) and monotonicity of $L$ we hence obtain

$$\mathbb{P}\left[ Y_{\text{test}} \notin C_{\hat{\lambda}}(X_{\text{test}}) \right] = \mathbb{E}\left[ L_{X_{\text{test}},Y_{\text{test}}}(\hat{\lambda}) \right] \leq \mathbb{E}\left[ L_{X_{\text{test}},Y_{\text{test}}}(\lambda^\dagger) \right] \leq \alpha$$

$\square$

## B DATASETS AND PRE-TRAINED MODELS

### B.1 DATA ACKNOWLEDGMENT

The authors acknowledge the National Cancer Institute and the Foundation for the National Institutes of Health, and their critical role in the creation of the free publicly available LIDC/IDRI Database used in this study.

### B.2 DATASETS

The M&Ms-2 dataset[1] Campello et al. (2021); Martín-Isla et al. (2023) comprises 360 patients with various pathologies affecting the right and left ventricles, as well as healthy subjects. For each patient, the dataset provides cardiac MRI images along with annotations for the left and right ventricles and the left ventricular myocardium. It includes both short-axis and long-axis MRI images; however, our experiments utilized only the short-axis images. We adhered to the predefined training and test splits. The training set was used for model training, while the original test set was further divided into two subsets: a calibration set and a reduced test set. The reduced test set included only a portion of the original test, i.e. the first 900 MRIs.

The MS-CMR19 dataset[2] Gao et al. (2023); Wu & Zhuang (2022); Zhuang (2018) is another cardiac dataset, but it includes different modalities. This variation introduces greater uncertainty in the predictions. The dataset features 45 patients and contains cardiac MR images taken from the short-axis view. In this instance, we also utilize pre-defined splits, extracting the calibration set from the original test set.

The third medical dataset[3] LIDCArmato III et al. (2015) (Licence CC BY 3.0) contains lungs CT images with the corresponding segmentations obtained across over 1000 patients. Two labels are annotated, namely background and cancer.

We created two separate datasets from the COCO dataset Lin et al. (2014). Specifically, we selected images that feature either animals or humans to form the COCO-animals dataset, and images that contain vehicles or humans to create the COCO-vehicles dataset. The COCO-animals dataset includes the following labels: background, cat, dog, sheep, cow, horse, bird, and human. In contrast, the COCO-vehicles dataset contains these labels: background, train, bus, bicycle, airplane, car, boat, and human. All images from both datasets have been used for calibration and testing, as we utilized a pre-trained model for this setup.

In Table 1 we provide the details regarding the datasets used in the experiment section.

---

[1]https://www.ub.edu/mnms-2/

[2]https://zmiclab.github.io/zxh/0/mscmrseg19/

[3]https://www.cancerimagingarchive.net/collection/lidc-idri/

Table 1: Summary of datasets

| Dataset | Calibration Images | Test Images | $L$ | Sampling Strategy |
|---|---|---|---|---|
| M&Ms-2 | 500 | 179 | 4 | Monte Carlo dropout |
| MS-CMR19 | 500 | 98 | 4 | Monte Carlo dropout |
| LIDC | 700 | 809 | 2 | Probabilistic U-Net |
| COCO an. | 275 | 39 | 8 | Ensemble Networks |
| COCO veh. | 275 | 46 | 8 | Ensemble Networks |

### B.3 PRE-TRAINED MODELS

For the two cardiac datasets we used a simple U-Net Ronneberger et al. (2015) trained with dropout. The architecture consists of an encoder-decoder structure with skip connections between corresponding levels to preserve spatial context. The encoder comprises a series of block modules, each with two convolutional layers followed by ReLU activations, batch normalization, and dropout for regularization. Feature maps are progressively downsampled using max pooling, doubling the number of channels at each depth. The decoder utilizes bilinear upsampling and 1×1 convolutions to reduce channel dimensions. At each stage of the decoder, the feature maps are concatenated with corresponding encoder outputs via skip connections to recover spatial resolution. The final output is produced through a 1×1 convolution to map to the desired number of segmentation labels. The U-Net model was trained using a learning rate of $3 \cdot 10^{-4}$, optimized via Adam Kingma & Ba (2017). The encoder network utilized an initial number of 48 filters, which doubled at each layer up to a fixed depth of 5. Input MRI scans were cropped to a spatial resolution of 128 × 128 pixels, with each pixel representing 1.375 mm in real-world space. Only MRIs with non-zero ground truth are used. A batch size of 2 was used , and the model was trained for 1500 epochs. A dropout rate of 0.4 was applied within encoder and decoder blocks (except at the final level of the encoder).

For the LIDC experiment we used a pytorch re-implementation of the probabilistic U-Net [4] Kohl et al. (2018). We trained the model with hyperparameters and splitting provided in the code. Both the original code[5] and the re-implementation are published under the Apache License Version 2.0.

Finally, for the COCO experiments we rely on an ensemble networks strategy based on DeepLabV3+ Chen et al. (2018; 2017). To generate different segmentation samples we used six different models with different backbones[6]: DeepLabV3-MobileNet, DeepLabV3-ResNet50, DeepLabV3-ResNet101, DeepLabV3Plus-MobileNet, DeepLabV3Plus-ResNet50, DeepLabV3Plus-ResNet101. The code is published under the MIT License.

### B.4 BASELINE METHODS

As described in the main text, the SACP method aggregate the score of neighborhood pixels. Let $\pi$ be a permutation of indices such that $f(X^{ij})_{\pi(1)} \geq \cdots \geq f(X^{ij})_{\pi(L)}$, then

$$S(X^{ij}, l) = \sum_{m=1}^{l} f(X^{ij})_{\pi(m)} + r(l),$$

$$S_{SACP}(X^{ij}, l) = (1 - w) \cdot S(X^{ij}, l) + \frac{w}{|N(X^{ij})|} \sum_{p \in N(X^{ij})} S(X^p, l),$$

$$\mathcal{T}^{SACP}(X^{ij}) = \{\pi(1), \ldots, \pi(k)\}, \ k = \min\left\{l \in \{1, \ldots, L\} : S_{SACP}(X^{ij}, l) > \lambda\right\}.$$

The hyper-parameter $w$ is the weight that regulate the strength of the aggregation, while $N(X^{ij})$ is a set that includes the neighborhood pixels. The dimension of this set can be also tuned, selecting how many pixels to consider for the aggregation. Then we can define the corresponding prediction sets

$$\mathcal{C}_\lambda^{SACP-}(X) = \left\{Y : \forall i, j \ Y^{ij} \in \mathcal{T}^{SACP}(X^{ij})\right\}, \quad \mathcal{C}_\lambda^{SACP}(X) = \left\{Y : \exists \tilde{Y} \in \mathcal{C}_\lambda^{PW-}(X) : Y \stackrel{\beta}{=} \tilde{Y}\right\}.$$

---

[4]https://github.com/stefanknegt/Probabilistic-Unet-Pytorch

[5]https://github.com/SimonKohl/probabilistic_unet

[6]https://github.com/VainF/DeepLabV3Plus-Pytorch

In Liu et al. (2025) they introduce an iterative score aggregation operator $\mathcal{V}$ as

$$\mathcal{V}_k(X^{ij}, l) = (1-w) \cdot \mathcal{V}_{k-1}(X^{ij}, l) + \frac{w}{|N(X^{ij})|} \sum_{p \in N(X^{ij})} \mathcal{V}_{k-1}(X^p, l),$$

where $\mathcal{V}_0 = S$. In our experiments, we keep the iterations equal to 1, since the over-smoothing of the scores lead to worst results.

## C  METRIC DETAILS

The Chao estimator is a commonly used non-parametric method in ecology and other fields for estimating the true species richness, or the total number of species, in a community based on sample data. This method addresses the challenge of unobserved species that may not be detected due to limited sampling efforts Chao (1984). It has been proven that the Chao estimator asymptotically converges to a lower bound of the true species richness as the sample size increases, i.e., as the number of observed individuals $S \to \infty$, the estimator converges to a consistent lower bound of the total number of species. The Chao estimator is not defined if $f_2 = 0$. In that case the following bias-corrected estimator needs to be used

$$\hat{N}_{CH} = S_1 + \frac{f_1(f_1 - 1)}{2(f_2 + 1)}.$$

The Pearson correlation $\rho_{i,j}$ between two vectors $\mathbf{y}_i$, $\mathbf{y}_j \in \mathbb{R}^n$ is computed using the standard formula

$$\rho_{ij}(\mathbf{y}_i, \mathbf{y}_j) = \frac{\sum_{n=1}^{N} (y_{i,n} - \bar{\mathbf{y}}_i)(y_{j,n} - \bar{\mathbf{y}}_j)}{\sqrt{\sum_{n=1}^{N}(y_{i,n} - \bar{\mathbf{y}}_i)^2} \sqrt{\sum_{n=1}^{N}(y_{j,n} - \bar{\mathbf{y}}_j)^2}}.$$

In order to seed up the computations, the Chao estimator and correlation have been computed considering only the non-constant pixels over the samples. It is clear that the results are equivalent to computing the metric considering the whole segmentation.

## D  IMPLEMENTATION DETAILS AND COMPUTATIONAL EXPENSES

We perform each experiment using a GPU NVIDIA A100-SXM4-40GB. For the optimization of the coefficient $\mathbf{c}$ we utilize an Adam optimizer with learning rate equal to 1 for the medical datasets and 10 for the COCO datasets. For every experiment in Section 3 we used $d\lambda = 0.01$ for both CONSIGN and the baselines. For the implementation of RAPS we chose $\theta = 0.05$ and $k_{reg} = \frac{L}{2}$, where $L$ is the number of labels. For the SACP we chose a neighborhood weight $w = 0.3$ and a neighborhood size of 7x7. The hyper-parameters were selected based on optimal performance.

The algorithm to numerically solve the optimization problem is described in Algorithm 2, while the pixel-wise/SACP calibration algorithm is described in Algorithm 3.

---

**Algorithm 2** Optimization algorithm approx_solver

    **Input:** $Y, \boldsymbol{\mu}(X), \{\mathbf{u}_k\}_{k=1}^K, lr, \mathcal{B}, T$
    **Output:** $\mathbf{c}^*$
1: optimizer $\leftarrow$ Adam($\mathbf{c}, lr$)
2: **for** $epoch \leftarrow 1$ **to** $T$ **do**
3:     $\boldsymbol{\sigma} \leftarrow \boldsymbol{\mu}(X) + \sum_{k=1}^K c_k \mathbf{u}_k$
4:     $loss \leftarrow \mathcal{L}(Y, P(\boldsymbol{\sigma}))$              $\triangleright$ with $\mathcal{L}$ as in equation 11
5:     $\mathbf{c} \leftarrow$ Adam.step()
6:     $\mathbf{c} \leftarrow proj_{\mathcal{B}}(\mathbf{c})$
7: $\mathbf{c}^* \leftarrow \mathbf{c}$

---

In Table 2, we compare the computational times of CONSIGN and the pixel-wise method. Notice that the computational time of SACP is equivalent to the pixel-wise one. The offline time is measured in minutes and considers an average calibration step for one calibration/test split. The online time is measured in seconds and refers to the sampling of $S$ segmentation from the prediction set. The

---

**Algorithm 3** Calibration algorithm for pixel-wise RAPS and SACP

> **Input:** $\alpha, \beta, d\lambda, \{(X_i, Y_i)\}_{i=1}^{N_{cal}}$
> **Output:** $\hat{\lambda}$

1: $\lambda \leftarrow 0; \hat{R} \leftarrow 1; \mathcal{I} \leftarrow \emptyset$
2: **while** $\hat{R} > \alpha - \frac{1-\alpha}{N_{cal}}$ **do**
3:      **for** $i \leftarrow 1$ **to** $N_{cal} \setminus \mathcal{I}$ **do**
4:          construct label set $\mathcal{T}^{PW/SACP}(X_i)$ as in equation 12
5:          **if** $Y_i \in \mathcal{C}_\lambda^{PW/SACP}(X_i)$ **then**    ▷ with $\mathcal{C}_\lambda^{PW/SACP}(X_i)$ as in equation 13/equation B.4
6:             $\mathcal{I} \leftarrow \mathcal{I} \bigcup \{i\}$
7:      $\hat{R} \leftarrow 1 - \frac{|\mathcal{I}|}{N_{cal}}$
8:      **if** $\hat{R} \leq \alpha - \frac{1-\alpha}{N_{cal}}$ **then**
9:          $\hat{\lambda} \leftarrow \lambda$
10:      **else**
11:          $\lambda \leftarrow \lambda + d\lambda$

Table 2: Comparison of offline and online times for CONSIGN and pixel-wise

| Method | Offline (min) | Online $S = 1$ (s) | Online $S = 10^3$ (s) | Online $S = 10^4$ (s) |
|---|---|---|---|---|
| CONSIGN | $\sim 5 - 15$ | $\sim 0.026 - 0.040$ | $\sim 0.4 - 3$ | $\sim 4 - 40$ |
| PW (RAPS) | $\sim 0.1 - 1$ | $\sim 0.2 - 0.5$ | $\sim 0.4 - 2$ | $\sim 3 - 15$ |

offline time is higher due to the SVD, but mostly due to the numerical solution of the minimization problem. However, the most important metric is the online time. Our method demonstrates faster online processing times for smaller sample sizes $S$. This is because we sample a vector in $\mathbf{c} \in \mathbb{R}^K$ instead of selecting a possible label for each pixel in the label set $\mathcal{T}^{PW}$. When we sample a large number of segmentations, the reconstruction process becomes more expensive, resulting in higher computational times. Nevertheless, our method maintains competitive efficiency overall, remaining fast even with larger sample sizes. The online computational time during the online phase includes Singular Value Decomposition (SVD), which adds only a constant time of approximately $2 - 20$ ms per image.

## E   ADDITIONAL EXPERIMENTS

In Figures 6-7-8-9-10-11-12-13-14-15 we show additional quantitative and qualitative result of our method. In Tables 3-4-5 we provide the calibrated $\hat{\lambda}$ for the experiments of Section 3 and further experiments.

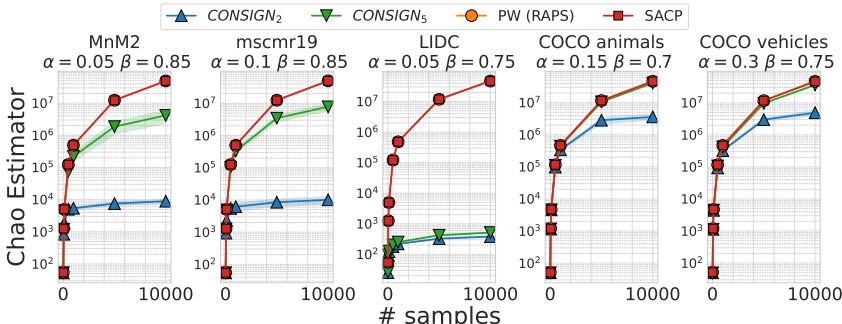

Figure 6: $\hat{N}_{CH}(\mathcal{Y}^*)$, $\hat{N}_{CH}(\mathcal{Y}^{PW})$ and $\hat{N}_{CH}(\mathcal{Y}^{SACP})$ for different experiments and principal components $K$

Table 3: Calibrated $\lambda$ across different experiments and splits for CONSIGN

| Dataset | $(\alpha, \beta, K)$ | $\hat{\lambda}$ **Fold 1** | $\hat{\lambda}$ **Fold 2** | $\hat{\lambda}$ **Fold 3** | $\hat{\lambda}$ **Fold 4** | $\hat{\lambda}$ **Fold 5** |
|---|---|---|---|---|---|---|
| M&Ms-2 | $(0.1, 0.9, 2)$ | 0.060 | 0.090 | 0.070 | 0.070 | 0.090 |
| | $(0.1, 0.9, 5)$ | 0.050 | 0.070 | 0.070 | 0.060 | 0.070 |
| | $(0.05, 0.85, 2)$ | 0.250 | 0.270 | 0.230 | 0.260 | 0.320 |
| | $(0.05, 0.85, 5)$ | 0.150 | 0.160 | 0.120 | 0.100 | 0.240 |
| | $(0.25, 0.95, 2)$ | 0.700 | 0.500 | 0.700 | 0.600 | 0.800 |
| | $(0.25, 0.95, 5)$ | 0.250 | 0.250 | 0.300 | 0.250 | 0.250 |
| | $(0.05, 0.9, 2)$ | 1.800 | 4.200 | 1.800 | 1.900 | 3.300 |
| | $(0.05, 0.9, 5)$ | 0.700 | 1.000 | 0.700 | 0.700 | 1.000 |
| MS-CMR19 | $(0.1, 0.85, 2)$ | 0.060 | 0.060 | 0.030 | 0.060 | 0.080 |
| | $(0.1, 0.85, 5)$ | 0.050 | 0.050 | 0.030 | 0.050 | 0.050 |
| | $(0.05, 0.8, 2)$ | 0.080 | 0.080 | 0.040 | 0.110 | 0.100 |
| | $(0.05, 0.8, 5)$ | 0.060 | 0.060 | 0.030 | 0.060 | 0.080 |
| | $(0.15, 0.9, 2)$ | 0.200 | 0.250 | 0.150 | 0.350 | 0.200 |
| | $(0.15, 0.9, 5)$ | 0.120 | 0.140 | 0.080 | 0.140 | 0.100 |
| | $(0.2, 0.9, 2)$ | 0.080 | 0.100 | 0.060 | 0.080 | 0.070 |
| | $(0.2, 0.9, 5)$ | 0.060 | 0.060 | 0.050 | 0.060 | 0.060 |
| LIDC | $(0.2, 0.8, 2)$ | 0.010 | 0.020 | 0.020 | 0.010 | 0.020 |
| | $(0.2, 0.8, 5)$ | 0.010 | 0.020 | 0.020 | 0.010 | 0.020 |
| | $(0.05, 0.75, 2)$ | 0.020 | 0.020 | 0.020 | 0.020 | 0.020 |
| | $(0.05, 0.75, 5)$ | 0.020 | 0.020 | 0.020 | 0.020 | 0.020 |
| | $(0.05, 0.85, 2)$ | 0.050 | 0.040 | 0.050 | 0.050 | 0.050 |
| | $(0.05, 0.85, 5)$ | 0.050 | 0.040 | 0.050 | 0.050 | 0.050 |
| | $(0.1, 0.9, 2)$ | 0.050 | 0.060 | 0.050 | 0.050 | 0.060 |
| | $(0.1, 0.9, 5)$ | 0.050 | 0.040 | 0.050 | 0.050 | 0.050 |
| COCO an. | $(0.25, 0.8, 2)$ | 0.030 | 0.030 | 0.030 | 0.040 | 0.040 |
| | $(0.25, 0.8, 5)$ | 0.020 | 0.020 | 0.020 | 0.020 | 0.030 |
| | $(0.15, 0.7, 2)$ | 0.060 | 0.060 | 0.070 | 0.060 | 0.180 |
| | $(0.15, 0.7, 5)$ | 0.040 | 0.040 | 0.050 | 0.040 | 0.050 |
| | $(0.2, 0.75, 2)$ | 0.020 | 0.040 | 0.040 | 0.040 | 0.080 |
| | $(0.2, 0.75, 5)$ | 0.010 | 0.020 | 0.020 | 0.020 | 0.030 |
| | $(0.2, 0.7, 2)$ | 0.01 | 0.01 | 0.01 | 0.01 | 0.02 |
| | $(0.2, 0.7, 5)$ | 0.01 | 0.01 | 0.01 | 0.01 | 0.01 |
| COCO veh. | $(0.3, 0.7, 2)$ | 0.110 | 0.030 | 0.060 | 0.060 | 0.030 |
| | $(0.3, 0.7, 5)$ | 0.060 | 0.020 | 0.030 | 0.030 | 0.020 |
| | $(0.3, 0.75, 2)$ | 0.330 | 0.300 | 0.300 | 0.300 | 0.300 |
| | $(0.3, 0.75, 5)$ | 0.120 | 0.100 | 0.110 | 0.110 | 0.100 |
| | $(0.25, 0.7, 2)$ | 0.500 | 0.300 | 0.300 | 0.300 | 0.300 |
| | $(0.25, 0.7, 5)$ | 0.150 | 0.100 | 0.100 | 0.150 | 0.100 |
| | $(0.35, 0.75, 2)$ | 0.100 | 0.050 | 0.050 | 0.100 | 0.050 |
| | $(0.35, 0.75, 5)$ | 0.060 | 0.030 | 0.030 | 0.040 | 0.020 |

Table 4: Calibrated $\lambda$ across different experiments and splits for pixel-wise (RAPS) method

| Dataset | $(\alpha, \beta)$ | $\hat{\lambda}$ **Fold 1** | $\hat{\lambda}$ **Fold 2** | $\hat{\lambda}$ **Fold 3** | $\hat{\lambda}$ **Fold 4** | $\hat{\lambda}$ **Fold 5** |
|---|---|---|---|---|---|---|
| M&Ms-2 | (0.1, 0.9) | 0.610 | 0.710 | 0.630 | 0.640 | 0.690 |
| | (0.05, 0.85) | 0.850 | 0.860 | 0.850 | 0.860 | 0.910 |
| | (0.25, 0.95) | 0.630 | 0.640 | 0.630 | 0.620 | 0.630 |
| | (0.05, 0.9) | 0.930 | 0.930 | 0.920 | 0.930 | 0.960 |
| MS-CMR19 | (0.1, 0.85) | 0.670 | 0.690 | 0.600 | 0.670 | 0.690 |
| | (0.05, 0.8) | 0.790 | 0.790 | 0.680 | 0.790 | 0.790 |
| | (0.15, 0.9) | 0.750 | 0.750 | 0.710 | 0.760 | 0.740 |
| | (0.2, 0.9) | 0.650 | 0.660 | 0.640 | 0.670 | 0.660 |
| LIDC | (0.2, 0.8) | 0.690 | 0.690 | 0.690 | 0.690 | 0.690 |
| | (0.05, 0.75) | 0.830 | 0.810 | 0.810 | 0.810 | 0.820 |
| | (0.05, 0.85) | 0.900 | 0.900 | 0.900 | 0.900 | 0.900 |
| | (0.1, 0.9) | 0.900 | 0.900 | 0.890 | 0.900 | 0.900 |
| COCO an. | (0.25, 0.8) | 0.560 | 0.560 | 0.570 | 0.590 | 0.610 |
| | (0.15, 0.7) | 0.710 | 0.710 | 0.710 | 0.710 | 0.720 |
| | (0.2, 0.75) | 0.590 | 0.610 | 0.620 | 0.620 | 0.670 |
| | (0.2, 0.7) | 0.520 | 0.510 | 0.540 | 0.540 | 0.580 |
| COCO veh. | (0.3, 0.7) | 0.710 | 0.670 | 0.680 | 0.680 | 0.640 |
| | (0.3, 0.75) | 0.810 | 0.750 | 0.760 | 0.780 | 0.740 |
| | (0.25, 0.7) | 0.820 | 0.800 | 0.800 | 0.800 | 0.800 |
| | (0.35, 0.75) | 0.700 | 0.660 | 0.680 | 0.680 | 0.620 |

Table 5: Calibrated $\lambda$ across different experiments and splits for SACP method

| Dataset | $(\alpha, \beta)$ | $\hat{\lambda}$ **Fold 1** | $\hat{\lambda}$ **Fold 2** | $\hat{\lambda}$ **Fold 3** | $\hat{\lambda}$ **Fold 4** | $\hat{\lambda}$ **Fold 5** |
|---|---|---|---|---|---|---|
| M&Ms-2 | (0.1, 0.9) | 0.580 | 0.680 | 0.600 | 0.620 | 0.670 |
| | (0.05, 0.85) | 0.820 | 0.850 | 0.820 | 0.820 | 0.870 |
| | (0.25, 0.95) | 0.600 | 0.610 | 0.610 | 0.600 | 0.600 |
| | (0.05, 0.9) | 0.870 | 0.890 | 0.880 | 0.880 | 0.920 |
| MS-CMR19 | (0.1, 0.85) | 0.660 | 0.660 | 0.600 | 0.660 | 0.660 |
| | (0.05, 0.8) | 0.770 | 0.770 | 0.680 | 0.770 | 0.770 |
| | (0.15, 0.9) | 0.710 | 0.720 | 0.670 | 0.720 | 0.710 |
| | (0.2, 0.9) | 0.630 | 0.640 | 0.610 | 0.650 | 0.630 |
| LIDC | (0.2, 0.8) | 0.710 | 0.720 | 0.720 | 0.710 | 0.720 |
| | (0.05, 0.75) | 0.830 | 0.820 | 0.820 | 0.820 | 0.840 |
| | (0.05, 0.85) | 0.900 | 0.900 | 0.890 | 0.900 | 0.900 |
| | (0.1, 0.9) | 0.890 | 0.890 | 0.890 | 0.890 | 0.890 |
| COCO an. | (0.25, 0.8) | 0.560 | 0.560 | 0.570 | 0.590 | 0.610 |
| | (0.15, 0.7) | 0.710 | 0.710 | 0.710 | 0.710 | 0.730 |
| | (0.2, 0.75) | 0.590 | 0.600 | 0.630 | 0.630 | 0.670 |
| | (0.2, 0.7) | 0.520 | 0.510 | 0.540 | 0.540 | 0.590 |
| COCO veh. | (0.3, 0.7) | 0.710 | 0.670 | 0.680 | 0.680 | 0.640 |
| | (0.3, 0.75) | 0.810 | 0.750 | 0.760 | 0.780 | 0.750 |
| | (0.25, 0.7) | 0.820 | 0.800 | 0.800 | 0.800 | 0.800 |
| | (0.35, 0.75) | 0.700 | 0.660 | 0.680 | 0.680 | 0.620 |

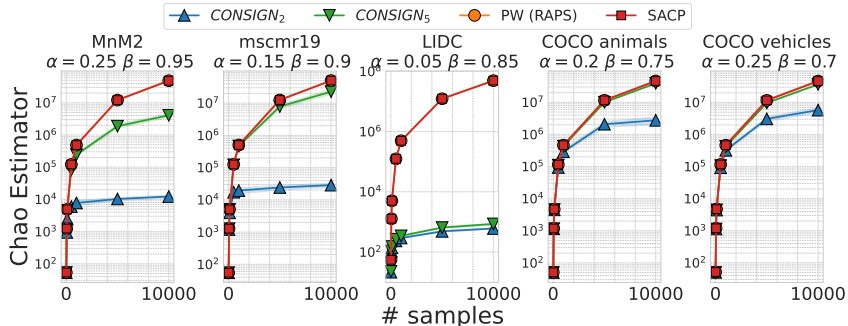

Figure 7: $\hat{N}_{CH}(\mathcal{Y}^*)$, $\hat{N}_{CH}(\mathcal{Y}^{PW})$ and $\hat{N}_{CH}(\mathcal{Y}^{SACP})$ for different experiments and principal components $K$

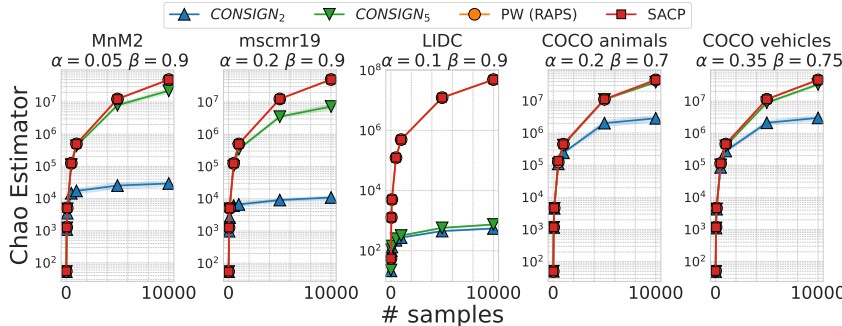

Figure 8: $\hat{N}_{CH}(\mathcal{Y}^*)$, $\hat{N}_{CH}(\mathcal{Y}^{PW})$ and $\hat{N}_{CH}(\mathcal{Y}^{SACP})$ for different experiments and principal components $K$

## F MISCELLANEOUS

In Figure 16, we show how CONSIGN can perform well even where the uncertainty is concentrated in small-scale details.

Finally, Figure 17 illustrates the relationship between the model predictions and the principal components derived from their SVD. For each experimental setting (as defined in Figure 5), we show the segmentation outputs of the pre-trained model $f$, along with the corresponding first four principal vectors $\mathbf{u}_i^0 \in \mathbb{R}^{H \times W}$ extracted from the label-0 channel. Recall that $\mathbf{u}_k \in \mathbb{R}^{L \times H \times W}$.

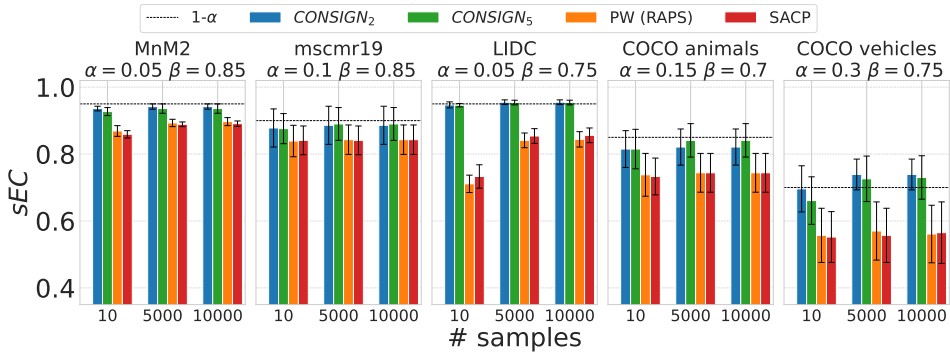

Figure 9: $sEC(\mathcal{Y}^*)$, $sEC(\mathcal{Y}^{PW})$ and $sEC(\mathcal{Y}^{SACP})$ for different experiments and principal components $K$

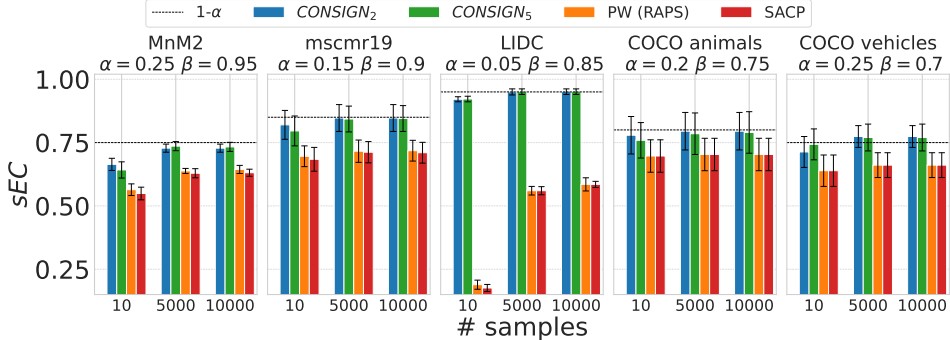

Figure 10: $sEC(\mathcal{Y}^*)$, $sEC(\mathcal{Y}^{PW})$ and $sEC(\mathcal{Y}^{SACP})$ for different experiments and principal components $K$

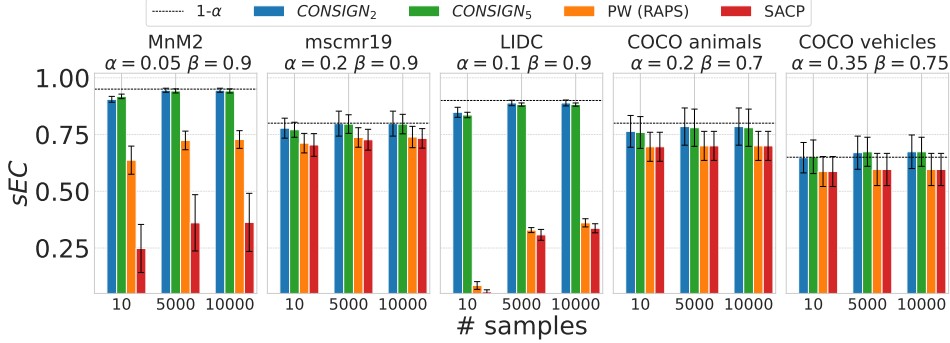

Figure 11: $sEC(\mathcal{Y}^*)$, $sEC(\mathcal{Y}^{PW})$ and $sEC(\mathcal{Y}^{SACP})$ for different experiments and principal components $K$

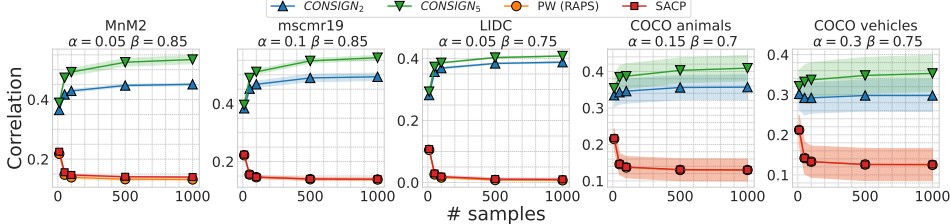

Figure 12: $\hat{\rho}(\mathcal{Y}^*)$, $\hat{\rho}(\mathcal{Y}^{PW})$ and $\hat{\rho}(\mathcal{Y}^{SACP})$ for different experiments and principal components $K$

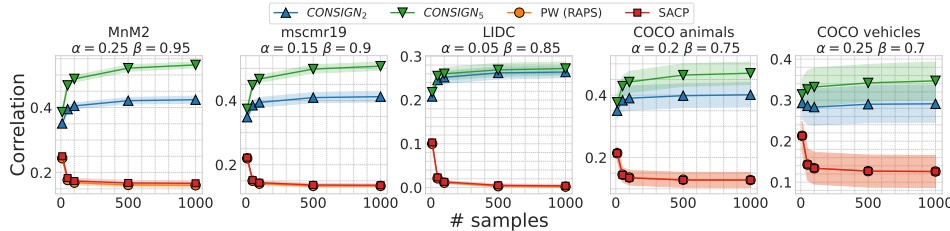

Figure 13: $\hat{\rho}(\mathcal{Y}^*)$, $\hat{\rho}(\mathcal{Y}^{PW})$ and $\hat{\rho}(\mathcal{Y}^{SACP})$ for different experiments and principal components $K$

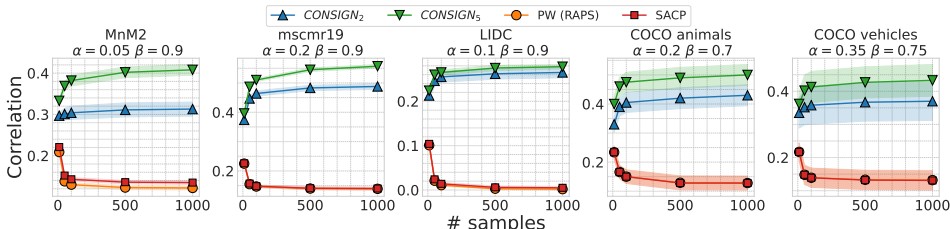

Figure 14: $\hat{\rho}(\mathcal{Y}^*)$, $\hat{\rho}(\mathcal{Y}^{PW})$ and $\hat{\rho}(\mathcal{Y}^{SACP})$ for different experiments and principal components $K$

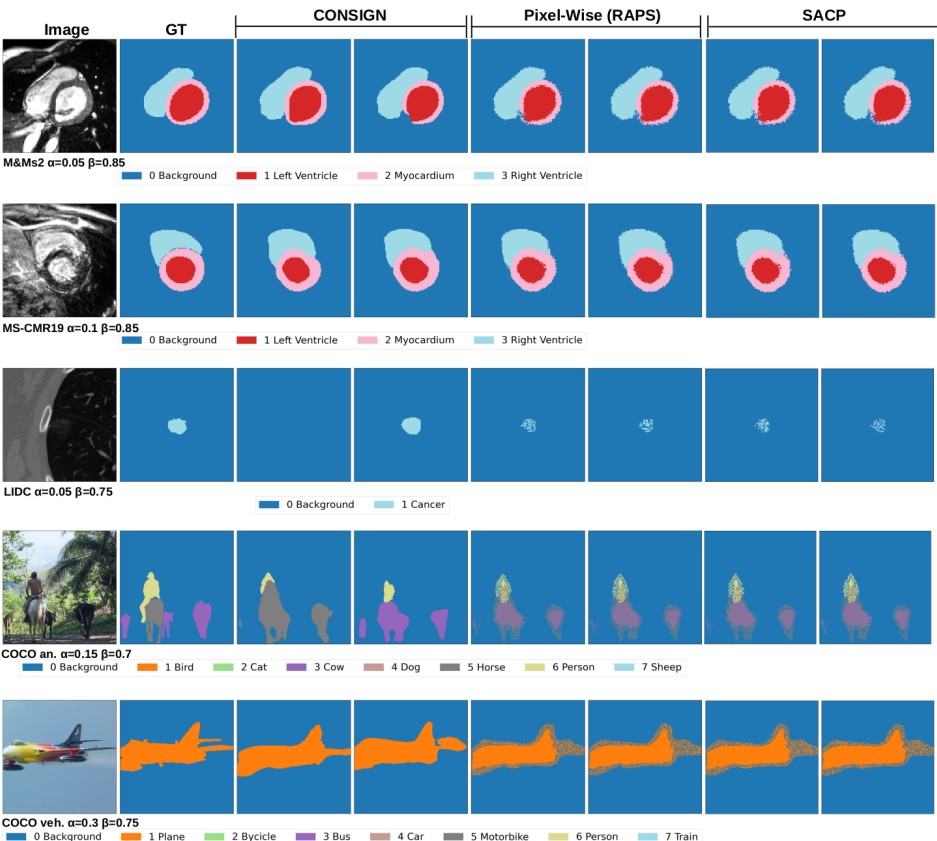

Figure 15: Qualitative comparison between samples from $\mathcal{Y}^*$ ($K = 5$), $\mathcal{Y}^{PW}$ and $\mathcal{Y}^{SACP}$.

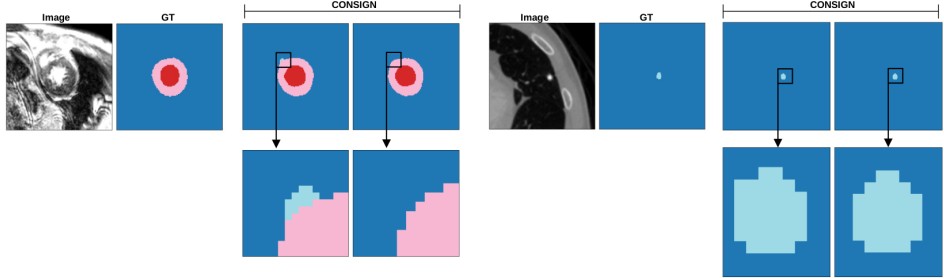

Figure 16: Samples from $\mathcal{Y}^*$ ($K = 5$) showing that CONSIGN is able to capture uncertainty also for small regions.

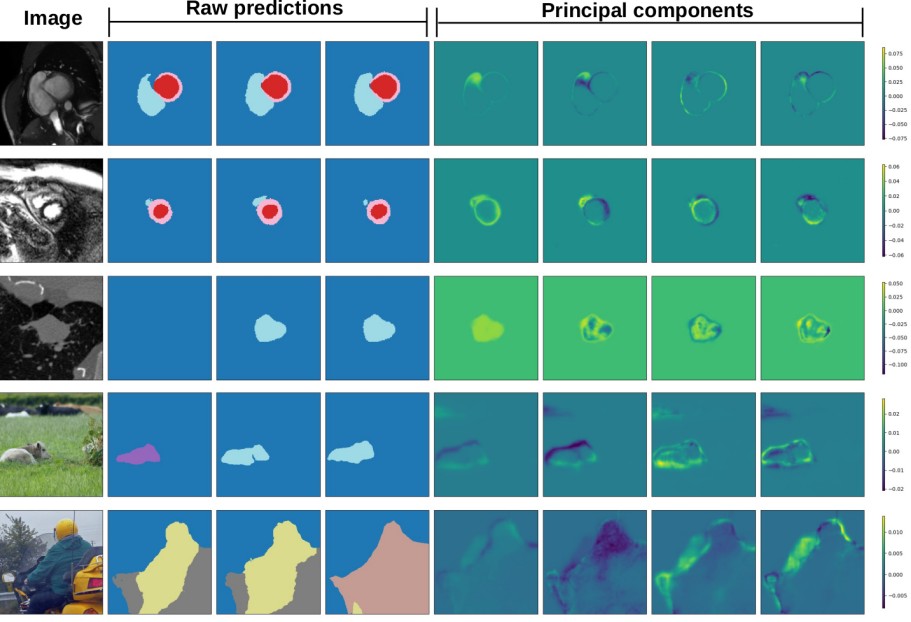

Figure 17: Example segmentation predictions produced by the pre-trained model $f$, and the corresponding first four principal vectors $\mathbf{u}_1^0, \mathbf{u}_2^0, \mathbf{u}_3^0, \mathbf{u}_4^0$ of the label-0 channel obtained from the SVD analysis. Each row corresponds to one of the experimental settings shown in Figure 5

