# OpenReview forum: "CONSIGN: Conformal Segmentation Informed by Spatial Groupings via Decomposition"
_ICLR.cc/2026/Conference — ICLR 2026 Poster_

### Official Review · Reviewer_cy3U · 2025-10-27

**Soundness:** 2
**Presentation:** 3
**Contribution:** 2
**Rating:** 4
**Confidence:** 5

**Summary:**

This paper introduces CONSIGN to improve uncertainty quantification in image segmentation. By leveraging spatial correlations between pixels, CONSIGN produces more meaningful prediction sets with rigorous coverage guarantees compared to traditional pixel-wise conformal prediction methods. The key innovation of CONSIGN is its ability to create spatially-aware prediction sets by incorporating principal component analysis to capture uncertainties in correlated regions, resulting in tighter, more interpretable prediction intervals. Through experiments on medical imaging datasets and COCO dataset subsets, the authors show that CONSIGN significantly outperforms existing CP-based methods, offering improved performance across multiple metrics, including uncertainty volume, empirical coverage, and correlation of samples.

**Strengths:**

1. CONSIGN effectively incorporates spatial correlations between pixels, a critical aspect in image segmentation tasks. This spatial awareness improves the quality of uncertainty estimates by reducing the size of prediction sets, providing more meaningful and interpretable results compared to traditional pixel-wise CP methods.
2. CONSIGN is highly adaptable to various applications and datasets without requiring significant changes to the underlying model architecture.

**Weaknesses:**

- The method assumes that the calibration and test datasets are exchangeable, which may not hold in many real-world scenarios where distribution shifts or out-of-distribution data are present. The current version of CONSIGN does not address such cases, and extending it to handle distribution shifts could significantly improve its robustness in dynamic environments.
- it heavily relies on the pre-trained model to generate multiple sample predictions. This means that the method's performance is closely tied to the quality of the pre-trained model, and it may not perform well (like inflated prediction sets) if the base model produces unreliable or poor-quality predictions
- The method excels at identifying larger uncertain regions, but for small, localized uncertainties (e.g., fine-grained segmentation boundaries), the spatial grouping approach might still lead to overly broad prediction sets. Fine-tuning how small-scale uncertainty is handled, or integrating multi-resolution strategies, could improve the method's accuracy in these cases.
- The calibration step involves solving a constrained minimization problem to identify the optimal parameter λ, this process can be numerically challenging and does not guarantee global optimality

**Questions:**

1. Could there be a more robust way to incorporate model uncertainty into the CONSIGN framework to mitigate potential performance degradation?
2. CONSIGN improves uncertainty quantification by leveraging spatial correlations, how does the method perform in regions with small or subtle uncertainties, such as fine boundaries in segmentation? Are the spatial correlations still effective in these contexts, or does the method tend to overestimate uncertainty in such cases?

---

> ### Author Response · Authors · 2025-11-19
>
> We thank the reviewer for carefully assessing our work and providing feedback. We provide a point-by-point response to the reviewers concerns and questions below.
> ***
> - **W1** We agree (and acknowledge in the limitations section) that this is an important limitation, shared by all standard conformal prediction methods, not unique to CONSIGN. Extending spatial-aware methods to handle distribution shifts is interesting future work, but these are orthogonal challenges. Our contribution is establishing spatial-aware conformal prediction; combining this with distribution adaptation is a natural next step.
> - **W2** We agree, but please note that this is true for all uncertainty quantification models. Our contribution is to provide a method for rigorous, spatially aware uncertainty bounds that can be applied on top of any suitable base model. If the base model is unreliable or produces poor-quality predictions, it is a feature of our method that the user is guaranteed to become aware of that by our method producing large prediction sets.
> - **W3** We believe that our experiments already suggest that our method is capable of identifying critical small uncertainty, especially in high-stake scenario like medical imaging. In order to further support hat, we have added additional examples in Figure 16 in the Appendix F of the revised version showing how fine-grained boundaries are handled.
> - **W4** We address this in Section 2.3: Even if we do not guarantee global optimality the statistical guarantee is **not** compromised. Failure to find the optimal $c$ only makes the algorithm more conservative (increases $\lambda$), which is safe. Moreover the optimization impacts only the calibration which is an offline procedure. The online procedure does not involve any numerical challenge.
> ***
> - **Q1** Even if we did not observe a significant performance degradation for small, localized uncertainties, we did observe that the performance of our method somehow correlates with meaningfulness of the samples provided by the base model. Thus, improved ways to incorporate model uncertainty into the CONSIGN framework are indeed desirable. We are currently working on further improving upon this by considering non-linear SVD approaches, which might be a promising direction for models that do not have a native way of providing sample predictions.
> - **Q2** Indeed, it seems that the spatial correlations are still effective in regions with small or subtle uncertainties; we have added additional experiments in Figure 16 of the Appendix F to further support this claim.

---

> > ### Comment · Reviewer_cy3U · 2025-11-20
> >
> > The responses have addressed my concerns, and I will raise my score. Thanks to the authors.

---

> > > ### Author Response · Authors · 2025-11-21
> > >
> > > We thank again the reviewer for the valuable feedback, which helped us improve and strengthen our work. We sincerely appreciate the time and effort put into reviewing our paper.

---

### Official Review · Reviewer_6bMr · 2025-10-27

**Soundness:** 3
**Presentation:** 3
**Contribution:** 3
**Rating:** 6
**Confidence:** 4

**Summary:**

The submission studies conformal prediction for segmentation models. The submission proposes to construct prediction sets that take into consideration the spatial correlation of the predictions of the base model in order to construct more informative prediction sets. The proposed method uses PCA to find principal directions along which to perturb the raw output of the model.

Experiments compare the proposed method with existing baselines on diverse datasets, containing both medical and natural images.

**Strengths:**

* Calibration of segmentation models is an important and under-developed field.
* Applying PCA decomposition is novel for conformal prediction of segmentation models.
* Experiments are comprehensive.

**Weaknesses:**

* Presentation could be made more precise.
* The proposed algorithm is not guaranteed to terminate.
* The construction is interesting, at the cost of less interpretable prediction sets.

**Questions:**

**Exit condition of Algorithm 1**

The main limitation of the current method is that it is not guaranteed to terminate with a finite parameter $\hat{\lambda} < \infty$. This seems to contradict the claim that the proposed method will eventually find a $\lambda$ that contains the ground truth.

- Could the authors expand on whether they experienced this behavior in their experiments?

- Could $\beta$ be chosen adaptively to guarantee termination of the algorithm?

**Experiments**

- Figure 1 should be introduced more clearly. It is only afterwards, in the experimental results, that it is explained how these segmentation masks are generated via sampling. To a first time reader, it is not clear how Figure 1 is generated.

- The fact that sampling from pixel-wise uncertainty intervals does not respect spatial correlations isn't particularly surprising on its own. Pixel-wise uncertainty intervals are not constructed to sample individual segmentation masks. So, it might be misleading to mention these as inherent limitations of existing works, as it is not what they are designed to solve.

- Have the authors considered applying PCA to the pixel-wise uncertainty intervals? This would also allow to sample segmentation masks that take into consideration the spatial correlation of the pixels. I wonder whether comparing with this baseline could provide stronger evidence in support of the proposed algorithm, that integrates spatial correlation at a deeper level.

- The name "sampled empirical coverage" might be confusing because it is reminiscent of the marginal statistical guarantee each method provides. That is, readers might expect all methods to provide sampled empirical coverage at level $1 - \alpha$ by design.

- It might be helpful to compare with Blot et al. "Automatically Adaptive Conformal Risk Control", who include examples on semantic segmentation.

---

**Minor comments**

- The operator $P$ is never defined. It is unclear what taking the argmax of a vector in $\mathbb{R}^{W \times H \times L}$ means. I assume $P$ first reshapes the column vector to $W \times H \times L$ and then takes the argmax along the $L$ dimension. This should be made clear.
- Is $\beta$ an error rate or an accuracy rate? Typo on line 206 $\geq 1 - \beta$?
- It might be helpful to explicitly define the indicator function in Eq. (8), or to spell it out as set inclusion.
- It might be helpful to include the raw predictions of the model in Fig. 5

---

> ### Author Response · Authors · 2025-11-19
>
> We thank the reviewer for the careful reading of our manuscript and for for the thoughtful and supportive feedback. We address first the weaknesses and then the questions.
> ***
> - **W1** We have improved the presentation in the revised version, incorporating all reviewers feedback. This includes expanding discussions of previously unclear sections, improving clarity in figure captions, conducting additional experiments and addressing minor typos
> - **W2**  Please refer to our answer of Question 1
> - **W3** It is true that our prediction sets are less interpretable in the sense that we cannot write them in an explicit form. We believe, however, that the elements of our prediction sets are much more interpretable compared to the pixel wise ones, see Figures 1-5
> ***
> - **Q1** Regarding the guaranteed existence of a finite $\lambda$, we refer to our response to Question 2 of reviewer jv9m. Regarding termination of the algorithm: It is true that we cannot exclude pathological cases in which the algorithm will not terminate. In our experiments we never experienced this behavior. But independent of this, it is important to note that we can always guarantee that the algorithm will **only** terminate with a prediction set with valid coverage guarantees. If it happens that the algorithm does not terminate (in reasonable time), it means the pre-trained model's performance is not sufficient to be the basis for a model with (reasonable) coverage guarantees. By using our algorithm for calibration, the user will be made aware of this; the algorithm will **not** silently return a prediction set for which the coverage guarantees fail. In the source code we have included a termination condition such that, after reaching a certain $\lambda_{\max}$, the algorithm will terminate, warning that only trivial prediction sets can be achieved for the model at hand. Also, we have added  the new Section 2.3.1 on this topic in the paper
> - **Q2** Implementation wise, this would be no problem. However, there is a statistical subtlety behind this question: While indeed a user could run our algorithm with different values of $\beta$ and would obtain the desired coverage guarantees for each $\beta$ separately, an algorithm that chooses $\beta$ adaptively depending on the calibration set would break the coverage guarantee: In this case, $\beta$ would be a function of the calibration data, a situation which is not covered by conformal risk control. Extending conformal risk control in this direction is an interesting research question, but would go beyond the scope of the present paper.
> - **Q3** We agree and have improved the caption of Figure 1 accordingly.
> - **Q4** We agree that pixel-wise uncertainty intervals are not constructed to sample individual segmentation masks. Our intention is to make explicit that the global uncertainty set defined by pixel-wise intervals implicitly contains all combinatorial label configurations. Sampling from this set is a way of presenting this prominently. But we agree that it might be misleading to mention this as inherent limitation of these works, and have added a clarifying sentence (line 454-455) and adapted the caption of Figure 1.
> - **Q5** Assuming that the suggestion is to sample from the calibrated pixel-wise uncertainty sets, and then applying PCA on these samples: We have tried it numerically, but believe one cannot get further insights from this. Since the original samples have no direct pixel-wise correlations, the resulting principle components do not show any meaningful spatial structure. More importantly, applying such an approach on top of the pixel-wise risk control would result in a method that does not have any rigorous uncertainty bounds.
> - **Q6** Thank you for pointing this out; we have renamed this metric to "Sample-based Estimated Coverage".
> - **Q7** Thank you for this reference, which is indeed relevant and has been added in the paper (lines 91-92). However, the main result of Blot et al. is to extend marginal conformal prediction towards conditional conformal prediction. In particular, they want to obtain coverage conditioned to different groups in the test set. The underlying conformal prediction method is still pixel-wise as are the baselines we compared to in the experiments. For these reasons, we  believe such a comparison would distract from the focus of our work. Techniques to extend conformal prediction from marginal towards conditional coverage is a separate research topic that is independent of adding spatial correlations to segmentation.
> - **Q8-11** We have fixed all of these: (1) explicitly define $P$ as the $\arg\max$ applied along the label dimension of the reshaped vector in $\mathbb{R}^{W \times H \times L}$ (line 197), (2) clarify $\beta$ is accuracy rate (line 215), therefore the $\ge\beta$ is correct (3) expand indicator function notation in Eq. 8. (line 227), (4) added raw prediction samples together with examples of SVD principal component in Figure 17 in appendix F.

---

> > ### Comment · Reviewer_6bMr · 2025-11-21
> > **Thank you for your response!**
> >
> > I sincerely thank the authors for their consideration of all reviewers comments.
> >
> > The authors have addressed all my comments and questions adequately in the revised version of the manuscript, and I will update my score accordingly.
> >
> > ---
> >
> > A follow-up comment on Q5:
> >
> > The baseline method I had in mind would first construct pixel-wise uncertainty intervals as existing methods do, and then perform PCA on the interval sizes, without sampling individual segmentation masks. This would be a first attempt at capturing spatial correlation. I agree, however, that statistical validity might be violated.

---

> > > ### Author Response · Authors · 2025-11-24
> > >
> > > We thank the reviewer once again for the thoughtful and constructive feedback throughout the revision process, and we also appreciate the clarification provided regarding Q5. We are grateful for the reviewer’s input and the time and effort dedicated to reviewing our paper

---

### Official Review · Reviewer_jv9m · 2025-10-30

**Soundness:** 3
**Presentation:** 3
**Contribution:** 3
**Rating:** 6
**Confidence:** 4

**Summary:**

The authors propose an approach, CONSIGN, within the conformal prediction framework to calibrate uncertainties output by an arbitrary probabilistic segmentation model. CONSIGN improves upon the existing literature in that it takes into account spatial correlations when outputting prediction sets with user-specified error guarantees. Prediction sets are derived from an SVD decomposition of the samples. The calibration involves an auxiliary constrained optimization step. The approach is demonstrated on 3 medical imaging datasets (M&Ms-2, LIDC and MS-CMR19) as well as two COCO subsets. The approach is tested on top of different segmentation frameworks across these applications: a dropout Unet, probabilistic Unet, and an ensemble networks strategy. The authors show qualitatively superior samples in the prediction sets compared to pixel-wise CP baselines and SACP, as well as report smaller prediction sets (in terms of Chao estimator and higher Pearson correlation between samples) at a fixed error-level, and better compliance to the specified error-level (sampled Empirical Coverage).

**Strengths:**

1. Significance and originality: The approach and the problem addressed are interesting, and this is ultimately what informed my choice of rating. There is potential for clear benefits from this type of approach as ignoring spatial correlations in segmentation uncertainties is problematic.

2. Qualitative and quantitative results over several datasets and segmentation backbones are promising.

**Weaknesses:**

1. It is not clear how alpha and beta are chosen across all applications (they differ for each application) and whether results generalize well to other values. Can the authors comment on these choices?

2. The method makes an i.i.d. assumption on the calibration and test sets. This is a strong assumption. In practice, guarantees of validity of the prediction sets may silently fail to apply for images with artefacts and other distribution shifts (unseen pathologies, different image quality, etc.). Yet these are exactly the cases that one would like to be flagged by UQ.

3. Overall readability/presentation could have been slightly improved (not a major concern however).

4. See questions and minor weaknesses below.

**Questions:**

5. It is not completely clear how Eq. 6 is derived and whether with this form, prediction sets have some guarantee of optimality. In particular as lambda is weighted by Sigma_k,k in Eq. 6, the range of values for A_k, B_k as a function of a_k,b_k is not completely clear. Sigma_k,k could also have appeared as a weight to c_k u_k(X) in the equation afterwards. Could the authors provide more details of derivations?

6. Can the authors justify the following claim : "In other words, even with a truncated PCA the prediction set will always include the ground truth, therefore we can use the standard CRC approach" ? I would assume that in some failure cases, none of the segmentation samples generated by the segmentation backbone will cover the ground truth; hence any prediction sets derived from them will in turn not cover the ground truth. Can this have substantial effects on the outcome of the proposed method in some degenerate cases?

7. Given that prediction sets for the proposed approach are derived with spatial awareness vs. pixel-wise for PW (RAPS), it is surprising that the estimated volume of prediction sets does not differ by more orders of magnitude in most experiments in Fig 2. Can the authors comment on this? The number of pixel-wise combinations should grow extremely fast compared to the number of spatially-consistent configurations.

---

> ### Author Response · Authors · 2025-11-19
>
> We thank the reviewer for their careful reading of our manuscript and for their positive comments and constructive suggestions. We are going to answer first to the weaknesses and then to the questions.
> ***
> - **W1** The choice of the parameters $\alpha$ (confidence level) and $\beta$ (accuracy rate) depends on application requirements and model quality: Regarding $\alpha$, one would choose smaller values (e.g., 0.05) for safety-critical applications and larger values (e.g., 0.3) for exploratory analysis. Similarly, $\beta$ defines an application specific pixel-wise accuracy rate: For applications where small-scale objects need to be segmented, $\beta$ should be high (e.g., 0.8), while a reliable UQ for segmenting larger objects can also be achieved with lower values. On the other hand, both $\alpha$ and $\beta$ define a model-specific balance between confidence/accuracy rate  and size of the prediction set: Well performing models $f$ allow for smaller choices of $\alpha $ and larger choices of $\beta$, while weaker models require more conservative choices in order to still achieve a meaningful prediction set. Based on this, we chose $\alpha$ and $\beta$ application-specific to provide a good balance between confidence level and size of the prediction set. We have added a discussion in Section 3.3 and additional experiments in Tables 3-5 and Figures 7-14 in Appendix E. Regarding generalization: Our results indicate that performance is consistent across a reasonable range for $\alpha$ and $\beta$.
> - **W2**  We agree that this is an important limitation, shared by all standard conformal prediction methods. Some recent works, such as Gibbs \& Candès (2021) (cited in Section 4), address this partially by considering temporal distribution shifts for scalar regression models (no segmentation). Combining spatial-aware methods with distribution-shift robustness is important future work for us, but these are orthogonal challenges: We see the contribution of this work in establishing the spatial-aware baseline for segmentation under standard CP assumptions
> - **W3**  We have improved the presentation in the revised version, incorporating all reviewers feedback. This includes expanding discussions of previously unclear sections, improving clarity in figure captions, conducting additional experiments and addressing minor typos
> ***
> - **Q1** We clarified this in the revised version (line 192): The derivation of Eq. (6) is heuristic, designed to obtain symmetric bounds around the empirical quantile midpoint that scale linearly with $\lambda$. The weighting by $\Sigma_{k,k}$ is natural, since principal components with higher variance get wider bounds, allowing more flexibility where the model shows more uncertainty. Directly putting  $\Sigma_{k,k}$ as a weight to the $c_k\mathbf{u}_k$ would have the undesired effect that the resulting overall bounds of the coefficient of $\mathbf{u}_k$ get shifted (it would result in intervals, with midpoint $c$ and boundaries $l$ and $r$, of the form $[\Sigma c - \lambda \Sigma l,\Sigma c + \lambda \Sigma r]$ instead of $[c - \lambda \Sigma l, c + \lambda \Sigma r]$).
> - **Q2**  We agree that the claim is imprecise and we have clarified it in the revised version (line 212-213): Our main point here is to state that, due to the non-linear projection $P$, even for small $K \in 2,5$, our method generally is able to produce prediction sets $\mathcal{C}_\lambda^*$ that eventually will contain the ground truth for large enough $\lambda$. This is in contrast to the regression setting, where one cannot expect to reach the ground truth even for arbitrary large $\lambda$. However, the reviewer is right that there might be some pathological cases (zero- or badly alligned entries of the principal directions $\mathbf{u}_k$) where our method can never reach the ground truth even for $\lambda \rightarrow \infty$. If this happens, our algorithm would not terminate with a finite $\lambda$, such that at least the user would be made aware of this and **there is no risk** of silently producing a prediction set that does not have the desired coverage guarantees. Please also refer to the answer to Question 1 of Reviewer 6bMr, to see how in practice we can terminate the calibration algorithm. Also, we have added the new Section 2.3.1 on this topic in the paper.
> - **Q3**  It also caught our attention that, while CONSIGN consistently outperforms the baselines in all cases, in some cases one would have expected an improvement in the estimated volume of prediction sets by even more orders of magnitude. Our explanation for this is that the performance of CONSIGN hinges on the meaningfulness of samples generated by the pre-trained model: With the probabilistic U-net, which was specifically trained to provide meaningful samples, our experiments show a difference by several orders of magnitude. In other cases, the difference to RAPS is less pronounced. We have added a comment in this direction in the paper (lines 410-413).

---

> > ### Comment · Reviewer_jv9m · 2025-11-21
> >
> > Thanks to the authors for adressing the reviewers' questions and incorporating the feedback in the paper. I will update my score accordingly.

---

> > > ### Author Response · Authors · 2025-11-24
> > >
> > > We would like to express our gratitude to the reviewer for the careful evaluation and constructive remarks. The feedback has been highly valuable in refining the manuscript, and we sincerely appreciate the time and attention dedicated to our work.

---

### Author Response · Authors · 2025-11-19
**General comment and code**

Thank you again to the AC and reviewers for the constructive feedback and for acknowledging the important strengths of our work — we truly appreciate the thoughtful evaluation. We have made the source code available in an anonymous repository, which can be accessed at the following link: https://mega.nz/folder/1VdTCa4Z#iiA_D3FT4vBJg--Hcv9e1g

---

### Author Response · Authors · 2025-12-01
**Summary Following Score Reversion**

For the benefit of the new Area Chair, we would like to highlight that the author–reviewer discussion was constructive and evidence-based. The exchange addressed all the reviewers’ concerns in detail, and **by the end of the discussion all reviewers expressed a well-justified, positive assessment of the paper.**

---

### Meta-Review · Area_Chair_gU75 · 2026-01-12

**Summary:**

This paper applies conformal prediction to segmentation problems, with the aim of obtaining statistically valid UQ. All reviewers agree that the paper should be published.

**Reviewer Concerns:**

The concerns were mainly clarifications. One important one regards the termination of the algorithm: it may not terminate, if the error level cannot be guaranteed. However, this did not happen in experiments.

**Reviewer Scores:**

All reviewers have indicated a willingness to accept the paper, and raise scores in accordance with this.

---

### Decision · Program_Chairs · 2026-01-26

Accept (Poster)